# Elevation-dependent trends in extreme snowfall in the French Alps from 1959 to 2019

Erwan Le Roux[1], Guillaume Evin[1], Nicolas Eckert[1], Juliette Blanchet[2], and Samuel Morin[3]

[1]Univ. Grenoble Alpes, INRAE, UR ETNA, Grenoble, France
[2]Univ. Grenoble Alpes, Grenoble INP, CNRS, IRD, IGE, Grenoble, France
[3]Univ. Grenoble Alpes, Univ. Toulouse, Météo France, CNRS, CNRM, CEN, Grenoble, France

**Correspondence:** Erwan Le Roux (erwan.le-roux@inrae.fr)

**Abstract.** Climate change projections indicate that extreme snowfall is expected to increase in cold areas, i.e. at high latitude and/or high elevation, and to decrease in warmer areas, i.e. at mid-latitude and low elevation. However, the magnitude of these contrasted patterns of change and their precise relations to elevation at the scale of a given mountain range remain ill-known. This study analyzes annual maxima of daily snowfall based on the SAFRAN reanalysis spanning the time period 1959-2019, and provided within 23 massifs in the French Alps every 300 m of elevation. We estimate temporal trends in 100-year return levels with non-stationary extreme value models that depend both on elevation and time. Specifically, for each massif and four elevation ranges (below 1000 m, 1000-2000 m, 2000-3000 m and above 3000 m), temporal trends are estimated with the best extreme value models selected on the basis of the Akaike information criterion. Our results show that a majority of trends are decreasing below 2000 m and increasing above 2000 m. Quantitatively, we find an increase of 100-year return levels between 1959 and 2019 equal to +23% (+32 kg m$^{-2}$) on average at 3500 m, and a decrease of -10% ($-7$ kg m$^{-2}$) on average at 500 m. However, for the four elevation ranges, we find both decreasing and increasing trends depending on location. In particular, we observe a spatially contrasted pattern, exemplified at 2500 m: 100-year return levels have decreased in the north of the French Alps while they have increased in the south which may result from interactions between the overall warming trend and circulation patterns. This study has implications for natural hazards management in mountain regions.

## 1 Introduction

Extreme snowfall can generate casualties and economic damages. For instance, it can cause major natural hazards (avalanche, winter storms) that might be intensified with high winds and freezing rain. Heavy snowfall can also disrupt transportation (road, rail and air traffic), tourism, electricity (power lines), and communication systems with a significant impact on economic services (Changnon, 2007; Blanchet et al., 2009). Subsequently, snow overloading can lead to the collapse of buildings such as a shed, a greenhouse, or as large as an exhibition hall (Strasser, 2008). It remains a counterintuitive phenomenon that extreme snowfall can increase in a warming climate, at least transiently, i.e. as long as local temperatures are cold enough (Frei et al., 2018). Therefore, to adapt protective measures, it is crucial to determine temporal trends in extreme snowfall for various areas (regions, elevations) and timescales, and to understand the underlying causes of these trends.

Extreme snowfall stems from extreme precipitation occurring in a range of optimal temperatures slightly below $0^oC$ according to O'Gorman (2014). This optimal range of temperatures both favours high precipitation intensities and percentages of precipitation falling as snow close to 100%. Thus, changes of extreme snowfall depend on a trade-off between trends in extreme precipitation and changes of the probability to experience temperature in this optimal range.

On a global scale, extreme precipitation is expected to increase with the augmentation of global mean temperature. Specifically, the most intense precipitation rates are theoretically expected to roughly increase at a rate of $7\%/^oC$, i.e. 7% per degree of global mean warming, due to an increase in maximum atmospheric water vapor content according to the Clausius–Clapeyron relationship (O'Gorman and Muller, 2010). In practice, the observed global mean temperature scaling for annual maxima of 1-day precipitation is $6.6\%/^oC$ (Sun et al., 2021). On the other hand, the probability to experience temperature in the optimal range for extreme snowfall is expected to decrease in warm areas, i.e. mid-latitude and low-elevation regions, as temperatures are expected to shift away from $0^oC$. However, this probability may increase in cold areas, i.e high-latitude and/or high-elevation regions, where temperatures are expected to shift toward $0^oC$ while remaining below $0^oC$ (Frei et al., 2018).

In the European Alps, past observations show both that the warming rate is larger than the global warming rate and that trends in extreme precipitation depend on the season and on the region. Indeed, past trends in mean annual surface temperature point to an increase in high mountain regions of Central Europe, with a warming rate ranging from 0.15 to $0.35^oC$ per decade since 1960 (Fig. 2.2 of IPCC 2019) against a range from 0.08 to $0.14^oC$ per decade since 1951 for the global warming rate (IPCC, 2013). Furthermore, past trends in daily maxima of precipitation largely depend on the season (Fig. 7 of Ménégoz et al. 2020). In winter, daily maxima precipitation (which may generate extreme snowfall) have trends that vary between -40% to +40% per century depending on the location. On the other hand, projected trends in winter precipitation in the European Alps indicate mostly positive trends in 100-year return levels (Fig. 12 of Rajczak and Schär 2017). For instance, an increase between 5% and 30% is expected under a high greenhouse gas emission scenario (comparing 2070–2099 to 1981–2010 for RCP8.5).

In and around the French Alps, studies analyzing extreme snowfall are rare (Beniston et al., 2018). Few papers describe the trends in extreme snowfall depending on elevation (Tab. 1). On one hand, past observations for Swiss stations below 1800 m present either a majority of decreasing trends or insignificant changes of the mean annual maximum of snowfall (Marty and Blanchet, 2012; Scherrer et al., 2013). On the other hand, climate projections for the Pyrenees and the Alps under a high greenhouse gas emission scenario (SRES A2 and RCP8.5, respectively) show that the mean seasonal maximum of snowfall is expected to decrease below a transition elevation and increase above it. López-Moreno et al. (2011) estimated a transition elevation around 2000 m for the Pyrenees (comparing 2070-2100 to 1960-1990), while Frei et al. (2018) estimated a transition around 3000 m for the Alps (comparing 2070–2099 to 1981-2010). In the French Alps, despite the existence of sufficient snowfall records and of previous studies having exploited them in an explicit extreme value framework, temporal trends in extreme snowfall remain poorly described. Indeed, earlier works rather focused on the spatial non-stationarity (w.r.t. latitude and longitude) of 3-day maxima of snowfall with max-stable processes (Davison et al., 2012). For instance, Gaume et al. (2013) estimated conditional 100-year return level maps at a fixed elevation of 2000 m, while Nicolet et al. (2016) found that the spatial dependence range of extreme snowfall has been decreasing.

| | Location | Indicator | Temporal trend | Time period | Dataset | Reference |
|---|---|---|---|---|---|---|
| **Past trends** | Switzerland | $S_{3d}$ | Decrease for the majority of stations | 1930-2010 | 25 stations, all except one below 1800 m | Marty and Blanchet (2012) |
| | Swiss Alps | $S_{1d}$ | Insignificant changes | 1864-2009 | 9 stations, all except one below 1000 m | Scherrer et al. (2013) |
| **Projected trends** | Pyrenees | 25-year return level of $S_{1d}$ | Decrease below 1500 m Increase above 2500 m | 1960-1990 vs. 2070-2100 | 1 HIRHAM RCM & SRES A2 | López-Moreno et al. (2011) |
| | Western and central Europe | Dec-Feb $S_{1d}$ | Increase almost only in high mountain ranges | 1961-2100 | 8 KNMI-RACMO2 RCM & RCP8.5 | de Vries et al. (2014) |
| | Alps | Sep-May $S_{1d}$ | Decrease below 3000 m Increase above 3000 m | 1981-2010 vs. 2070–2099 | 14 EURO-CORDEX GCM-RCM & RCP8.5 | Frei et al. (2018) |

**Table 1.** Temporal trends in extreme snowfall w.r.t. elevation in and around the French Alps. Elevations are in meters (m) above sea level. $S_{Nd}$ denotes the annual maximum of snowfall in N consecutive days.

This study addresses the gap identified above, by assessing past temporal trends in the 23 massifs of the French Alps, with special emphasis on the 100-year return levels of daily snowfall. We rely on the SAFRAN reanalysis (Durand et al., 2009) available for the period 1959-2019, which provides, among others, time series of daily snowfall (from which annual maxima can be computed) for each massif and every 300 m of elevation between 600 m and 3600 m (Vernay et al., 2019). In order to properly account for the specific statistical nature of maximal daily snowfall, our methodology relies on non-stationary extreme value models that depend both on elevation and time. Specifically, for each massif and four ranges of elevations (below 1000 m, 1000-2000 m, 2000-3000 m and above 3000 m), temporal trends in 100-year return levels are estimated with a model selected on the basis of the Akaike information criterion.

## 2 Snowfall data

We study annual maxima of daily snowfall in the French Alps, which are located between Lake Geneva to the north and the Mediterranean Sea to the south (Fig. 1). This region is typically divided into 23 mountain massifs of about 1000 km$^2$, which correspond to 23 spatial units covering the French Alps (Vernay et al., 2019), the climate being considered as homogeneous inside each massif for a given elevation.

The SAFRAN reanalysis (Durand et al., 2009; Vernay et al., 2019) combines large scale reanalyses and forecasts with in situ meteorological observations to provide daily snowfall data, i.e. snow water equivalent of solid precipitation measured in kg m$^{-2}$, available for each massif from August 1958 to July 2019. We consider annual maxima of daily snowfall centred on the winter season, e.g. an annual maximum for the year 1959 corresponds to the a maximum from the 1st of August 1958 to the 31st of July 1959. Thus, we study annual maxima from 1959 to 2019.

The SAFRAN reanalysis focuses on the elevation dependency of meteorological conditions. Indeed, this reanalysis is not produced on a regular grid, but provides data for each massif every 300 m of elevation. As illustrated in Figure 1, we consider four ranges of elevation: below 1000 m, between 1000 m and 2000 m, between 2000 m and 3000 m, and above 3000 m. For instance, the maxima for the range "below 1000 m" correspond to the maxima at 600 m and the maxima at 900 m. We note that for each massif, we do not have any maxima above the top elevation of the massif.

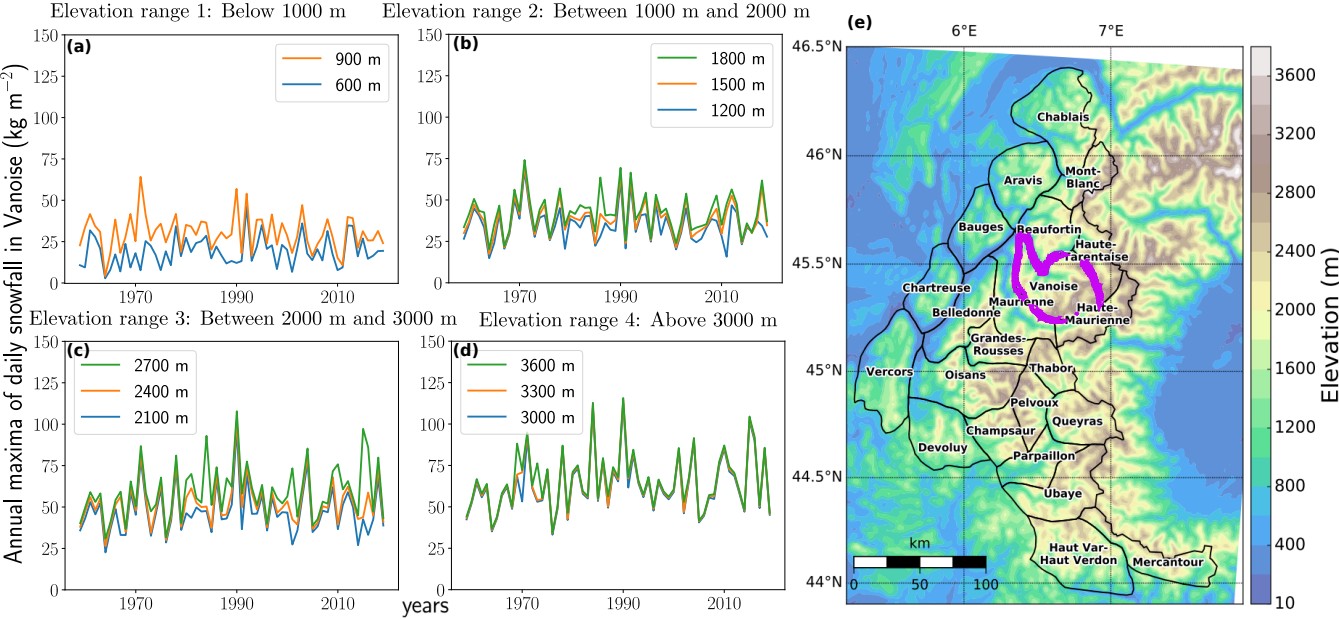

**Figure 1. (a, b, c, d)** Time series of annual maxima of daily snowfall from 1959 to 2019 for the Vanoise massif (purple region) clustered by the four ranges of elevations considered. **(e)** Topography and delineation of the 23 massifs of the French Alps (Durand et al., 2009).

The SAFRAN reanalysis has been both evaluated directly with in situ temperature and precipitation observations and indirectly with various snow depth observations compared to snow cover simulations of the model Crocus driven by SAFRAN atmospheric data (Durand et al., 2009; Vionnet et al., 2016; Quéno et al., 2016; Revuelto et al., 2018; Vionnet et al., 2019). Specifically, in Vionnet et al. (2019), the SAFRAN reanalysis has been evaluated for snowfall against two numerical weather prediction (NWP) systems for the winter 2011-2012. They find that the seasonal snowfall averaged over all the massifs of the French Alps reaches 546 mm in SAFRAN, 684 mm in the first NWP, and 737 mm in the second NWP. In details, they find that SAFRAN significantly differ from the two NWP systems in (i) areas of high elevation, probably due to the limited number of high-elevation stations and gauge undercatch (ii) on the windward side of the different mountain ranges due to the assumption of climatological homogeneity within each SAFRAN massif. In Ménégoz et al. (2020), the SAFRAN reanalysis has been compared to the regional climate model MAR which uses ERA-20C as forcing. They found that the vertical gradient of annual mean of total precipitation of SAFRAN is generally smaller than those simulated by MAR.

## 3  Method

### 3.1  Statistical distribution for annual maxima

Following the block maxima approach from extreme value theory (Coles, 2001), we model annual maxima of daily snowfall
with the generalized extreme value (GEV) distribution. Indeed theoretically, as the central limit theorem motivates asymptoti-
cally sample means modelling with the normal distribution, the Fisher–Tippett–Gnedenko theorem (Fisher and Tippett, 1928;
Gnedenko, 1943) encourages asymptotically sample maxima modelling with the GEV distribution. In practice, if $Y$ is a random
variable representing an annual maximum, we can assume that $Y \sim \text{GEV}(\mu, \sigma, \xi)$. Then, if $y$ denotes an annual maximum:

$$P(Y \leq y) = \begin{cases} \exp\left[-(1 + \xi \frac{y-\mu}{\sigma})_+^{-\frac{1}{\xi}}\right] \text{ if } \xi \neq 0 \text{ and where } u_+ \text{ denotes } \max(u, 0), \\ \exp\left[-\exp\left(-\frac{y-\mu}{\sigma}\right)\right] \text{ if } \xi = 0, \text{ in other words } Y \sim \text{Gumbel}(\mu, \sigma), \end{cases} \tag{1}$$

where the three parameters are: the location $\mu$, the scale $\sigma > 0$, and the shape $\xi$. The GEV distribution encompasses three
sub-families of distribution called reversed Weibull, Gumbel and Fréchet, which correspond to $\xi < 0, \xi = 0, \xi > 0$, respectively.

### 3.2  Elevational-temporal models

We consider non-stationary models that depend both on elevation and time. Such models combine a stationary random com-
ponent (a fixed extreme value distribution, e.g. GEV distribution) with non-stationary deterministic functions that map each
covariate to the changing parameters of the distribution (Montanari and Koutsoyiannis, 2014). Specifically, for each massif and
each range of elevation (below 1000 m, 1000-2000 m, 2000-3000 m and above 3000 m), if $Y_{z,t}$ represents an annual maximum
at the elevation $z$ (within one of the four ranges of elevation) for the year $t$ (between 1959 and 2019), we assume that:

$$Y_{z,t} \sim \text{GEV}(\mu(z,t), \sigma(z,t), \xi(z)). \tag{2}$$

As illustrated in Table 2, we consider eight models that verify Equation (2). For a model $\mathcal{M}$, we denote as $\boldsymbol{\theta}_{\mathcal{M}}$ the set of
parameters of $\mu(z,t), \sigma(z,t)$ and $\xi(z)$. Following a preliminary analysis with pointwise distributions (Sect. 4.1), we consider
models with a location and a scale parameters that vary linearly w.r.t. elevation. Then, the shape parameter is either constant or
linear w.r.t. elevation. Finally, the location and/or the scale can vary linearly with time. As shown in Table 2, we assume that
the temporal and elevational effects are separable inside each range of elevation. Thus, we do not consider models with cross
terms, i.e. terms involving both the elevation and the years such as $z \times t$. We discuss this assumption in Sect. 5.1.

First, models are fitted with the maximum likelihood method. Let $\boldsymbol{y} = (y_{z_1,t_1}, ..., y_{z_1,t_M}, ..., y_{z_N,t_1}, ..., y_{z_N,t_M})$ represents a
vector of annual maxima from year $t_1$ to $t_M$ and for the range of elevations containing $z_1, ..., z_N$ for a given massif (Sect. 2).
We classically assume that maxima are conditionally independent given $\boldsymbol{\theta}_{\mathcal{M}}$. For each model $\mathcal{M}$, we compute the maximum
likelihood estimator $\widehat{\boldsymbol{\theta}}_{\mathcal{M}}$ which corresponds to the parameter $\boldsymbol{\theta}_{\mathcal{M}}$ that maximizes the likelihood $p(\boldsymbol{y}|\boldsymbol{\theta}_{\mathcal{M}})$, where $p(\boldsymbol{y}|\boldsymbol{\theta}_{\mathcal{M}}) = \prod_z \prod_t \frac{\partial P(Y_{z,t} \leq y_{z,t})}{\partial y_{z,t}}$.

| Temporal stationarity | Model name | $\mu(z,t)$ | $\sigma(z,t)$ | $\xi(z)$ | # $\boldsymbol{\theta_{\mathcal{M}}}$ |
|---|---|---|---|---|---|
| Stationary | $\mathcal{M}_0$ | $\mu_0 + \mu_z \times z$ | $\sigma_0 + \sigma_z \times z$ | $\xi_0$ | 5 |
| | $\mathcal{M}_{\xi_z}$ | | | $\xi_0 + \xi_z \times z$ | 6 |
| Non-stationary | $\mathcal{M}_{\mu_t}$ | $\mu_0 + \mu_z \times z + \mu_t \times t$ | $\sigma_0 + \sigma_z \times z$ | $\xi_0$ | 6 |
| | $\mathcal{M}_{\mu_t,\xi_z}$ | | | $\xi_0 + \xi_z \times z$ | 7 |
| Non-stationary | $\mathcal{M}_{\sigma_t}$ | $\mu_0 + \mu_z \times z$ | $\sigma_0 + \sigma_z \times z + \sigma_t \times t$ | $\xi_0$ | 6 |
| | $\mathcal{M}_{\sigma_t,\xi_z}$ | | | $\xi_0 + \xi_z \times z$ | 7 |
| Non-stationary | $\mathcal{M}_{\mu_t,\sigma_t}$ | $\mu_0 + \mu_z \times z + \mu_t \times t$ | $\sigma_0 + \sigma_z \times z + \sigma_t \times t$ | $\xi_0$ | 7 |
| | $\mathcal{M}_{\mu_t,\sigma_t,\xi_z}$ | | | $\xi_0 + \xi_z \times z$ | 8 |

**Table 2.** Elevational-temporal models considered rely on the GEV distribution. For the elevational non-stationarity, the location and the scale parameters vary linearly with the elevation $z$, while the shape is either constant or linear with $z$. For temporal non-stationary models, the location and/or the scale vary linearly with time $t$.

Then, we select the model with the minimal Akaike information criterion (AIC) for each massif and range of elevations. Indeed, AIC is the best criterion in a non-stationary context with small sample sizes (Kim et al., 2017). AIC equals $2 \times [\# \boldsymbol{\theta_{\mathcal{M}}} - p(\boldsymbol{y}|\widehat{\boldsymbol{\theta}}_{\mathcal{M}})]$ where $\# \boldsymbol{\theta_{\mathcal{M}}}$ is the number of parameters for the model $\mathcal{M}$. Thus, minimizing the AIC corresponds to selecting models that both have few parameters, i.e. low $\# \boldsymbol{\theta_{\mathcal{M}}}$, and that fit well the data, i.e. high $p(\boldsymbol{y}|\widehat{\boldsymbol{\theta}}_{\mathcal{M}})$. Goodness-of-fit is assessed with Q-Q plots which show a good fit for the selected models (Appendix A).

## 3.3 Return levels

The $T$-year return level, which corresponds to a quantile exceeded each year with probability $p = \frac{1}{T}$ is the classical metric to quantify hazards of extreme events (Coles, 2001; Cooley, 2012). We set $p = \frac{1}{100} = 0.01$ as it corresponds to the 100-year return period which is widely used for hazard mapping and the design of defense structure in France, notably for snow-related hazards (Eckert et al., 2010). Let $\mathcal{M}$ denotes a model from Table 2, and $\widehat{\boldsymbol{\theta}}_{\mathcal{M}}$ the corresponding maximum likelihood estimator. Then, the associated return levels $y_p$, which depends on the elevation $z$ and the year $t$, can be computed as follows:

$$P(Y_{z,t} \leq y_p(z,t)|\widehat{\boldsymbol{\theta}}_{\mathcal{M}}) = 1 - p \leftrightarrow y_p(z,t) = \mu(z,t) - \frac{\sigma(z,t)}{\xi(z)}[1 - (-\log(1-p))^{-\xi(z)}]. \quad (3)$$

We study trends in return levels. For any considered model, the time derivative of the return level $\frac{\partial y_p(z,t)}{\partial t}$ is constant and quantifies the yearly change of return level. Thus, for each range of elevations, a massif is said to have an increasing trend if the associated return level has increased, i.e. if $\frac{\partial y_{0.01}(z,t)}{\partial t} > 0$. A massif has a decreasing trend if $\frac{\partial y_{0.01}(z,t)}{\partial t} < 0$. In the result section, we display changes of 100-year return levels between 1959 and 2019, i.e. over the last 60 years, which equals $60 \times \frac{\partial y_{0.01}(z,t)}{\partial t}$. If $\frac{\partial y_{0.01}(z,t)}{\partial t} \neq 0$, i.e. if the selected model is temporally non-stationary, we compute the significance of the trend with a semi-parametric bootstrap resampling approach (Appendix B). We generate $B = 1000$ bootstrap samples using the parameter $\widehat{\boldsymbol{\theta}}_{\mathcal{M}}$. For each bootstrap sample $i$, we compute the time derivative of the return level $\frac{\partial y_p(z,t)}{\partial t}^{(i)}$. Finally, a massif

with an increasing trend is said to have a significant trend if $\hat{p}(\frac{\partial y_{0.01}(z,t)}{\partial t} > 0|\widehat{\boldsymbol{\theta}}_{\mathcal{M}}) = \frac{1}{B}\sum_{i=1}^{B}\mathbb{1}_{\frac{\partial y_{0.01}(z,t)}{\partial t}(i) > 0} > 1 - \alpha$, where $\alpha = 5\%$ is the significance level. In other words, it is significantly increasing when the percentage of bootstrap samples for which the return levels are increasing is above the threshold $1 - \alpha$. Likewise, a massif has a significant decreasing trend if $\hat{p}(\frac{\partial y_{0.01}(z,t)}{\partial t} < 0|\widehat{\boldsymbol{\theta}}_{\mathcal{M}}) > 1 - \alpha$.

### 3.4 Workflow

In Section 4.1, we analyse changes of pointwise distribution of annual snowfall maxima with elevation in the 23 massifs of the French Alps, which helped us define the eight elevational-temporal models considered (Sect. 3.2). Pointwise distribution stands for a distribution fitted on the annual maxima from a single elevation of one massif. Specifically, we fit a pointwise GEV distribution with the maximum likelihood method for each massif every 300 m of elevation from 600 m to 3600 m. We exclude physical implausible distributions, i.e. $\xi \notin [-0.5, 0.5]$ (Martins and Stedinger, 2000). Then, we compute elevation gradients for the three GEV parameters and the 100-year return level with a linear regression.

In Section 4.2, we compare pointwise distributions with our approach based on piecewise elevational-temporal models. Piecewise models stands for models fitted on the annual maxima from a range of elevation of one massif. We present the elevational-temporal models selected in each massif for each range of elevations (below 1000 m, 1000-2000 m, 2000-3000 m and above 3000 m) obtained with the methodology described in Section 3.2. First, we fit the eight models from Table 2 with the maximum likelihood method. Then, we select one model with the AIC. Finally, if the selected model is temporally non-stationary, we assess the significance of the trend using a semi-parametric bootstrap resampling approach with a significance level $\alpha = 5\%$ (Sect. 3.3) .

In Section 4.3, we present for each massif and range of elevations the temporal trends in 100-year return levels obtained from selected models. We compute 100-year return levels in 2019 and its changes between 1959 and 2019 with the selected elevational-temporal models (Sect. 3.3). For each range of elevations, a massif has an increasing (resp. decreasing) trend if the associated return level has increased (resp. decreased).

### 4 Result

#### 4.1 Pointwise distribution for each elevation.

According to pointwise fits, the location and scale parameters increase linearly with elevation (Fig. 2 **a, b**). $R^2$ coefficients are always larger than 0.8, except for the Bauges massif for the scale parameter (Fig. 3 **b**). The average elevation gradient for the location and the scale parameters are equal to 2.1 kg m$^{-2}$/100 m and 0.39 kg m$^{-2}$/100 m, respectively (Fig. 3 **a, b**). In particular, this linear augmentation is also valid for any range of elevations considered to fit elevational-temporal models. Thus, as explained in Section 3.2, we assume for elevational-temporal models location and scale parameters that vary linearly w.r.t. elevation. On the other hand, changes of the shape parameter rarely follow a linear relationship with elevation between 600 m and 3600 m. Indeed, only seven massifs have $R^2$ coefficients larger than 0.5 (Fig. 3 **c**). However, as illustrated in Figure 2 **(c)**,

it does not preclude the shape parameter to vary linearly with the elevation locally, i.e. within an elevation range. Therefore, as explained in Section 3.2, we assume for elevational-temporal models that the shape parameter is either constant or linear w.r.t. elevation.

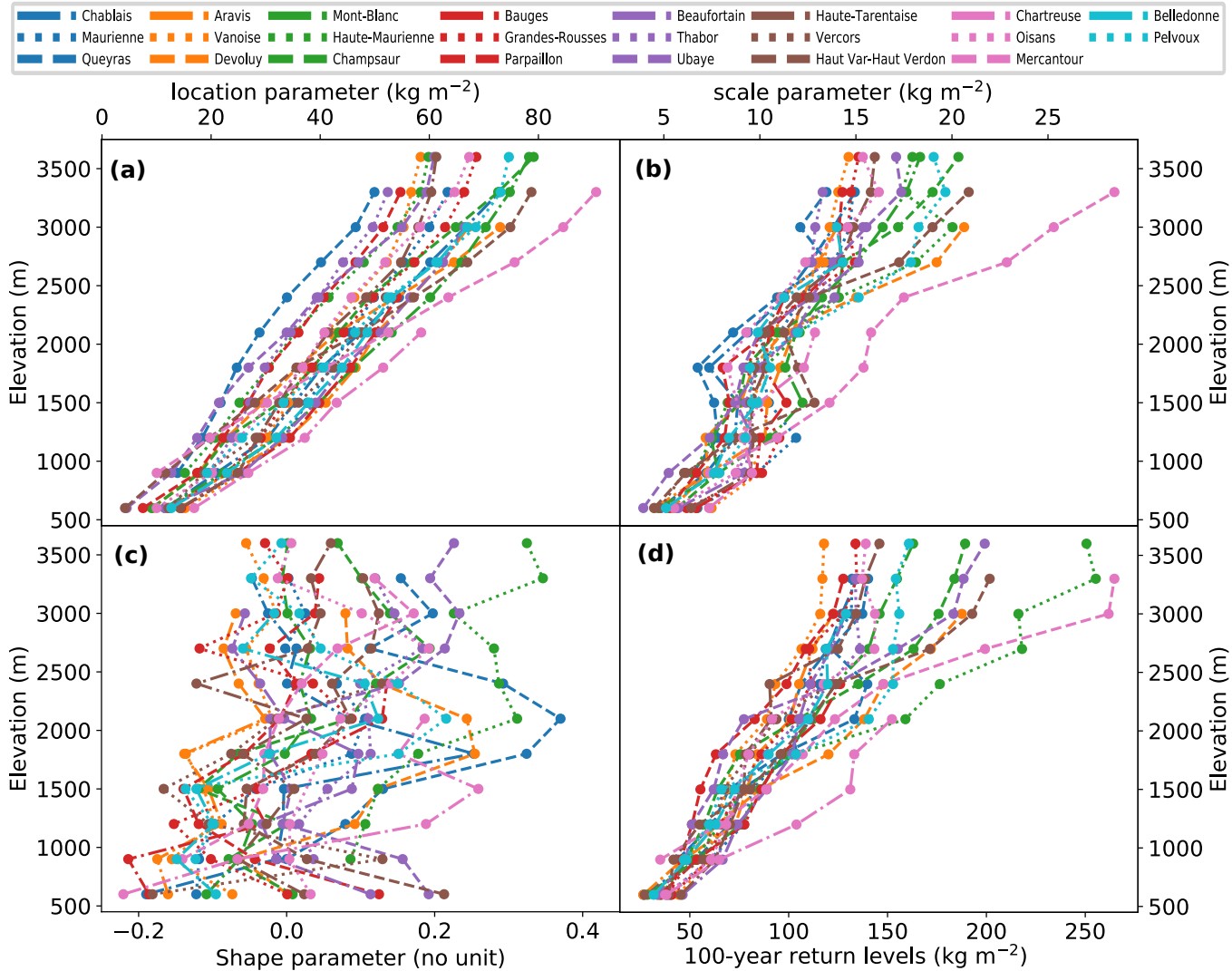

**Figure 2.** Changes of GEV parameters **(a, b, c)** and of 100-year return levels **(d)** with the elevation for the 23 massifs of the French Alps. GEV distributions are estimated pointwise for the annual maxima of daily snowfall every 300 m of elevation.

     In Figure 2 **d** we show the change of 100-year return levels with elevation, while in Figure 3 **d** we display their elevation gradients. Return levels augment linearly with elevation, which is confirmed by $R^2$ coefficients always larger than $0.8$. The largest

return levels and elevation gradients correspond to the Mercantour (Southern Alps) and Haute-Maurienne massif (eastern part of the French Alps).

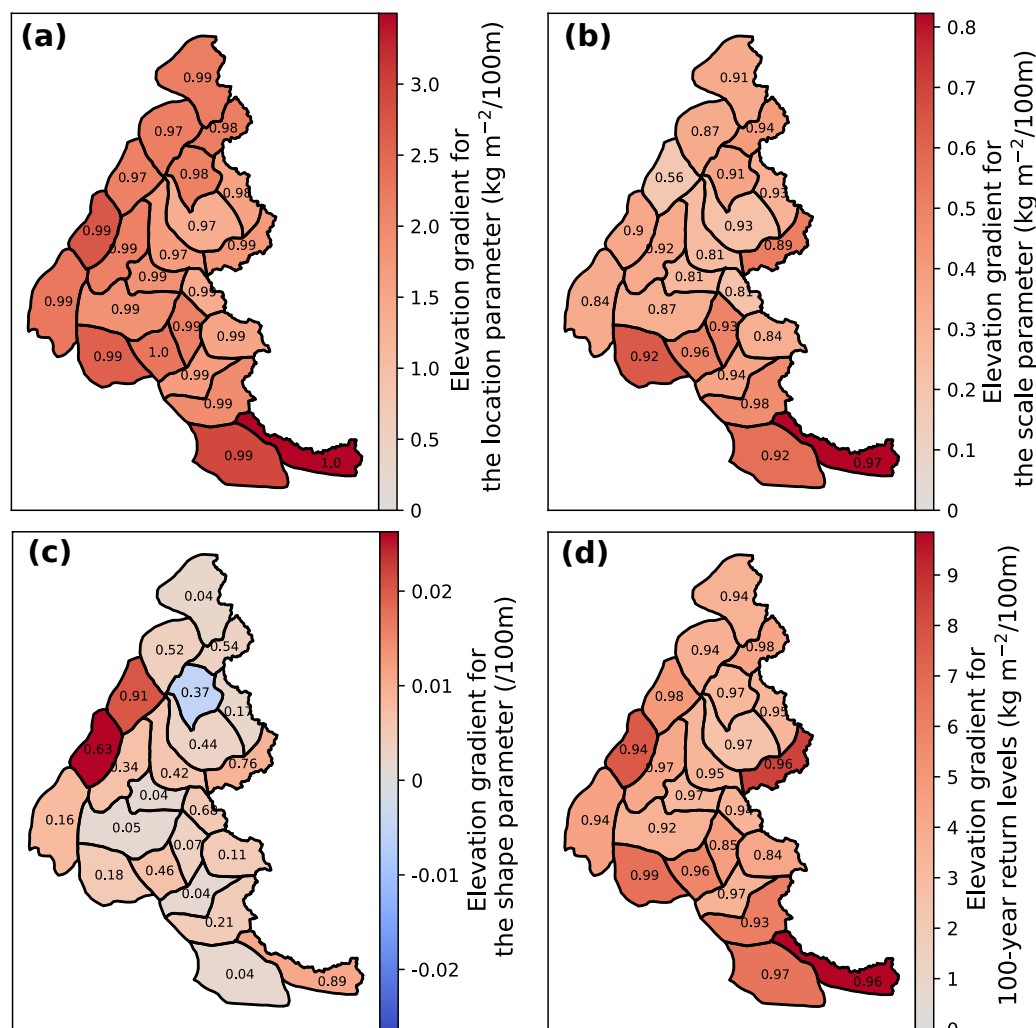

**Figure 3.** Elevation gradients for the GEV parameters **(a, b, c)** and 100-year return levels **(d)** for the 23 massifs of the French Alps. GEV distributions are estimated pointwise for the annual maxima of daily snowfall every 300 m of elevation. Elevation gradients are estimated with a linear regression. The $R^2$ coefficient is written in black for each massif.

## 4.2 Elevational-temporal models for each range of elevations

Figure 4 illustrates selected models for each massif and each range of elevations. The most selected model is a temporally non-stationary model $\mathcal{M}_{\mu_t,\sigma_t}$, which is selected for $54\%$ of the massifs, and is significant for $32\%$ of the massifs, respectively. Further, the temporally stationary model $\mathcal{M}_0$ and $\mathcal{M}_{\xi_z}$ have been selected for $28\%$ and $1\%$ of the massifs. The remaining temporally non-stationary models have been selected for $17\%$ of massifs. We notice that the four most represented temporally

non-stationary models have a linearity w.r.t. the years $t$ for the scale parameter (which impacts both the mean and variance of the GEV distribution), potentially indicating that often both the intensity and the variance of maxima are changing over time.

Furthermore, we observe that the shape parameter values remain between $-0.4$ and $0.4$, which is a physically acceptable range (Martins and Stedinger, 2000). Except a majority of Weibull distributions (massifs displayed in green in Figure 4) at 500 m in the northwest of the French Alps, and a clear majority of Fréchet distributions (yellow massifs) at 2500 m, we do not find any clear spatial/elevation patterns for the shape parameter.

Figure 5 exemplifies the differences between pointwise distribution and our approach based on piecewise elevational-temporal models. We consider annual maxima from 600 m to 3600 m of elevation for the Vanoise massif (Sect. 2). First, our approach makes it possible to interpolate GEV parameter values (and thus to deduce 100-year return levels) for each range of elevations (blue line) rather than having point estimate (green dot). Furthermore, it reduces confidence intervals for return levels (shaded areas) which were computed with an approach based on semi-parametric bootstrap resampling (Appendix B). Finally, our approach accounts for temporal trends. For example, for the Vanoise massif above 3000 m, the selected model is a temporally non-stationary model (Fig. 4) with an increasing trend in return levels (Fig. 8). This explains why return levels in 2019 estimated from the elevational-temporal model exceed return levels estimated from the pointwise distribution.

### 4.3 Temporal trends in return levels

Figure 6 shows that both increasing and decreasing trends in 100-year return levels are found for all elevation ranges. We also observe that a majority of trends are decreasing below 2000 m and increasing above 2000 m. If we analyse only significant trends the elevation pattern remains the same. On one hand, more than 30% of massifs are significantly decreasing below 2000 m: 40% below 1000 m, and 30% for the range 1000-2000 m. On the other hand, roughly 30% of massifs are significantly increasing above 2000 m: slightly less than 30% for the range 2000-3000 m, and slightly less than 40% above 3000 m. We note that the sign and the significance of the trends (summarized with the percentages on Fig. 6) remain largely similar for the trends in 10-year and 50-year return periods events (Appendix C).

In Figure 7, we show distributions of changes and relative changes of 100-year return levels between 1959 and 2019. We find a temporal increase of 100-year return levels between 1959 and 2019 equal to +23% (+32 kg m$^{-2}$) on average at 3500 m, and a decrease of -10% ($-7$ kg m$^{-2}$) on average at 500 m. For intermediate elevations, i.e. between 1000 m and 3000 m, we observe that the distribution of changes and of relative changes, remain roughly negative (decrease) at 1500 m, and positive (increase) at 2500 m. This result holds for all massifs, for the subset of massifs with a selected model temporally non-stationary, and for the subset with a selected model temporally non-stationary and significant.

In Figure 8, we display the change of 100-year return levels between 1959 and 2019 for each range of elevations. At 500 m, we observe that eight massifs have a stationary trend, and five massifs located in the center of the French Alps have a significant decreasing trend. We also note that two massifs located in the Western French Alps have an increasing significant trend, with an absolute change of 100-year return level close to +20 kg m$^{-2}$. At 1500 m, six massifs in the center of the French Alps have decreasing trends. At 2500 m, we observe a spatially contrasted pattern: most decreasing trends are located in the north, while

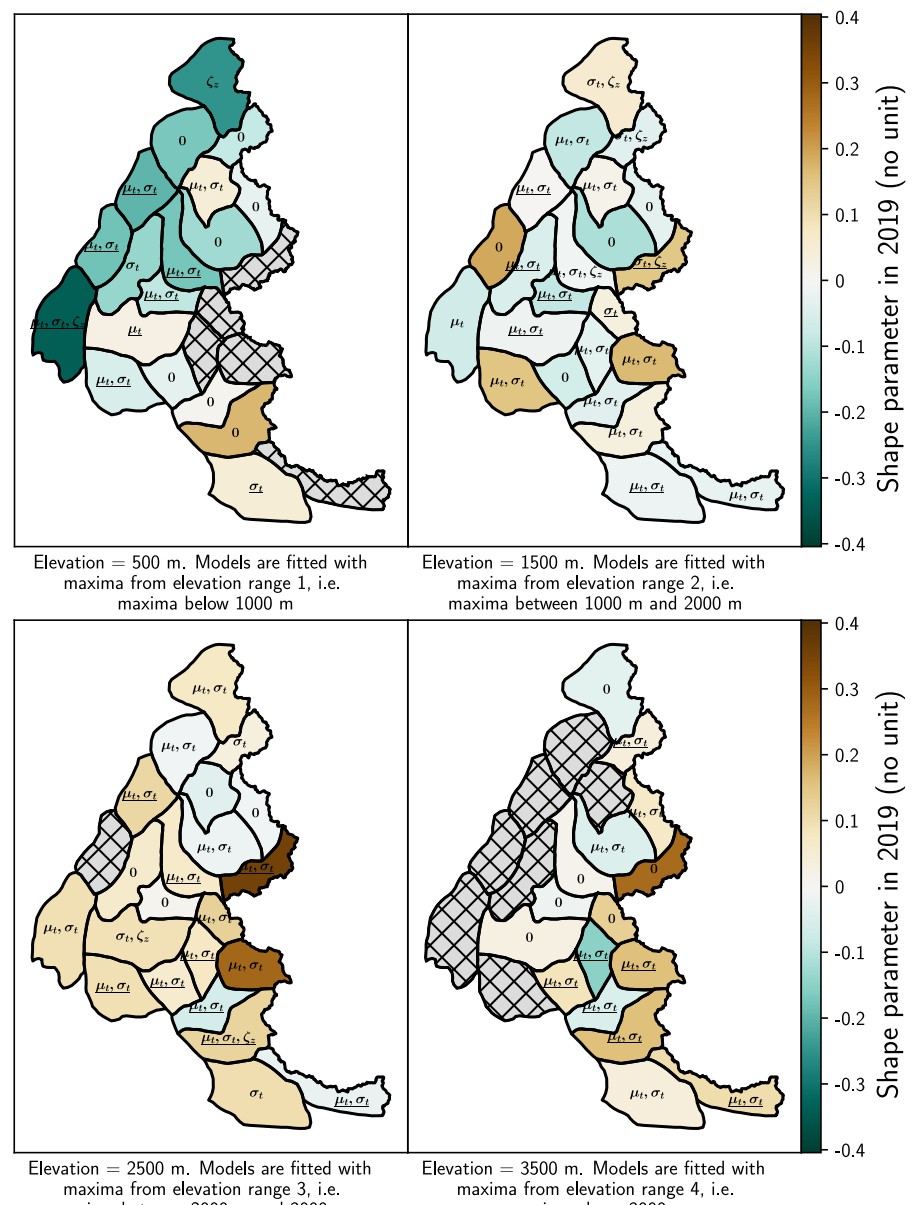

**Figure 4.** Selected models and shape parameter values for each range of elevations in the 23 massifs of the French Alps. We write the suffix of the name of each selected model on the map, e.g. we write $\mu_t, \sigma_t$ for the model $\mathcal{M}_{\mu_t, \sigma_t}$. We underline the suffix when the model has a significant trend (Sect. 3.3). Hatched grey areas denote missing data, e.g. when the elevation is above the top elevation of the massif. Shape parameter values are computed at the middle elevation for each range, e.g. at 1500 m for the range 1000-2000 m.

most increasing trends are located in the south. We discuss this pattern in Sect. 5.4. At 3000 m, we observe that the 6 massifs with a significant increasing trend are located in the South of the French Alps.

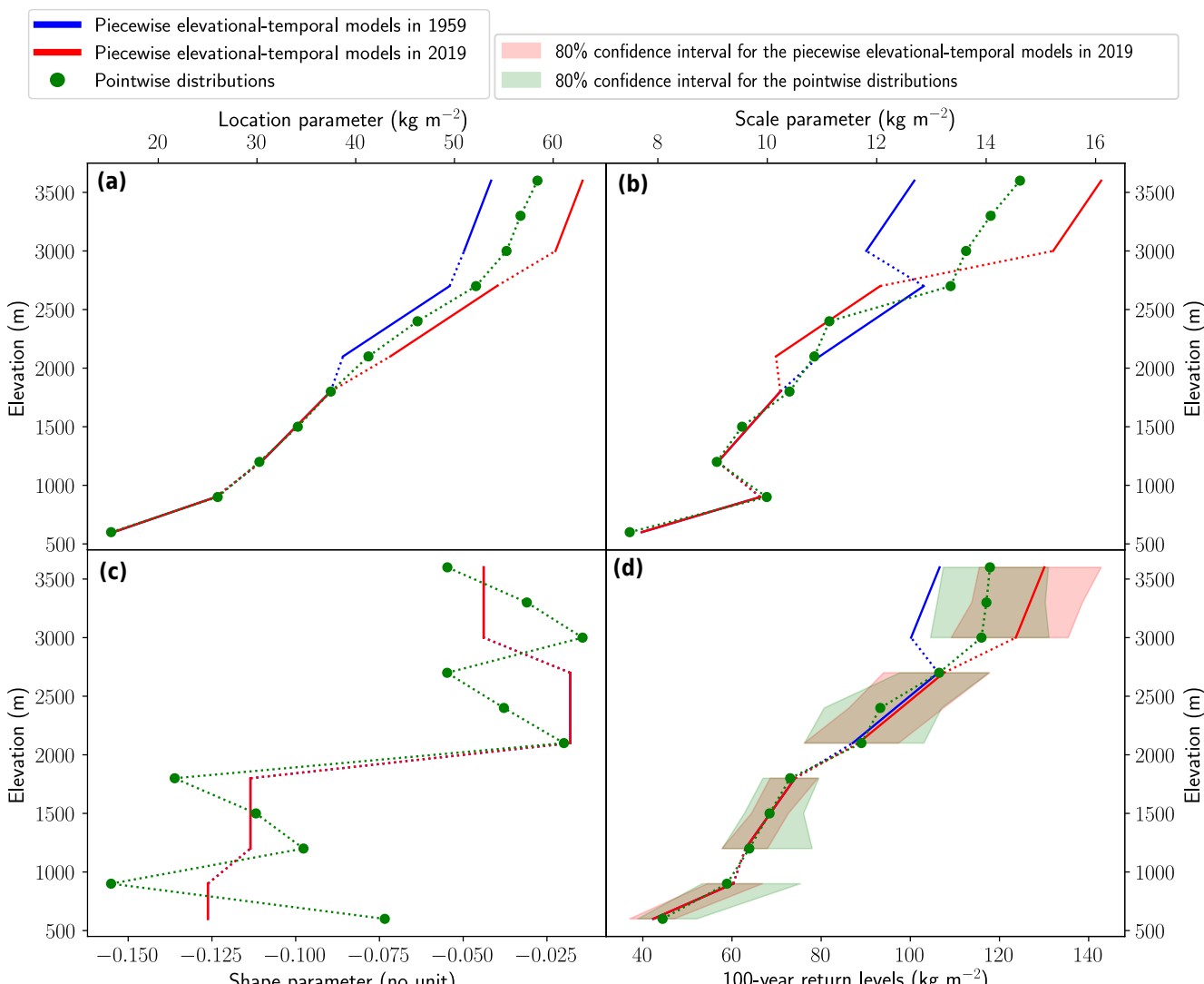

**Figure 5.** Comparison of pointwise distributions with our approach based on piecewise elevational-temporal models for the Vanoise massif. GEV parameters **(a, b, c)** and 100-year return levels with their 80% confidence intervals **(d)** are shown from 600 m to 3600 m of elevation.

In Figure 9, we display the return level in 2019 for each range of elevations. Combining Figures 8 and 9 enables us to pinpoint massifs with both high return levels in 2019 and with an increasing or decreasing trend. For instance, at 2500 m, the Haute-Maurienne massif has one of the highest 100-year return levels in 2019 (185 kg m$^{-2}$), but it is decreasing with time. On the other hand, at 2500 m and 3500 m, most massifs in the south have high 100-year return levels values in 2019 and increasing trends.

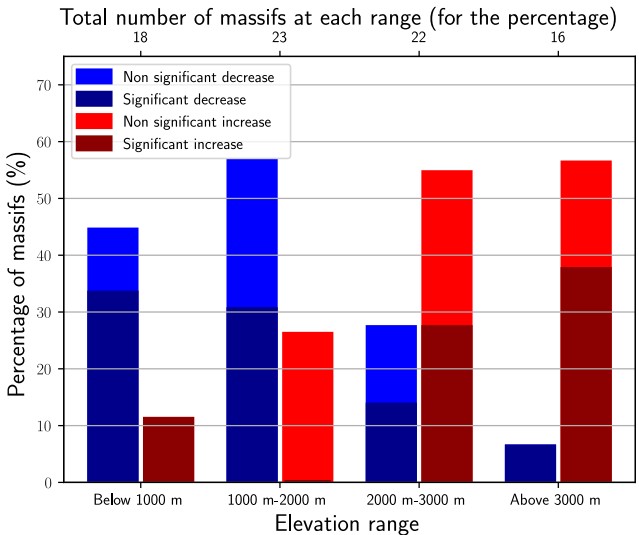

**Figure 6.** Percentages of massifs with significant/non-significant trends in 100-year return levels of daily snowfall for each range of elevation. A massif has an increasing/decreasing trend if the 100-year return level of the selected elevational-temporal model has increased/decreased.

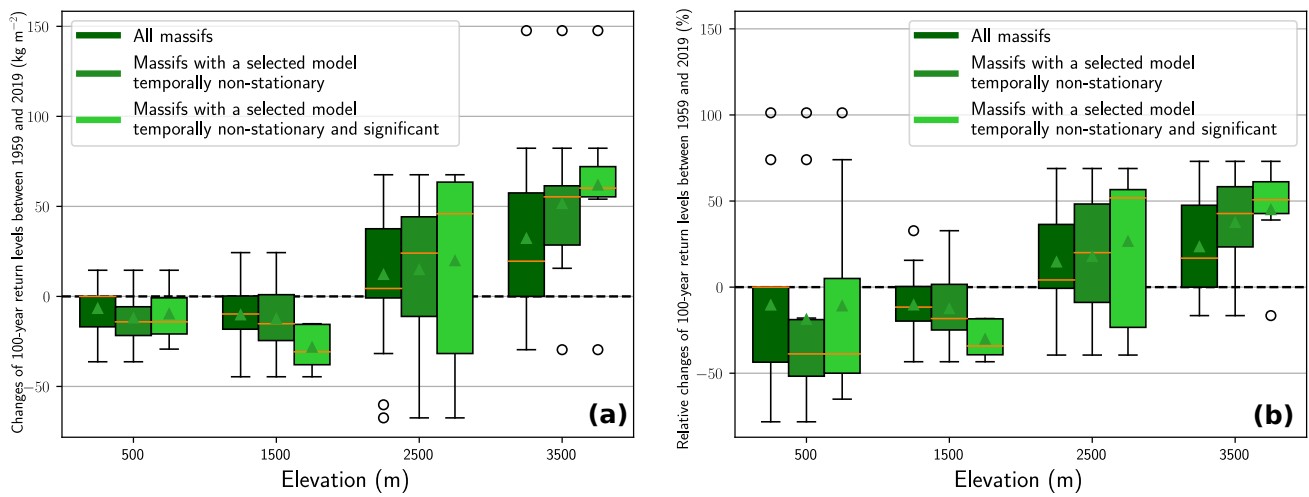

**Figure 7. (a)** Distributions of changes of 100-year return levels between 1959 and 2019 for one elevation in each range of elevation. The mean and the median are displayed with a green triangle and an orange line, respectively. **(b)** Same as **(a)** but for the relative changes. Distributions of changes are computed at the middle elevation for each range, e.g. at 1500 m for the range 1000-2000 m.

## 5 Discussion.

### 5.1 Methodological considerations

We discuss the statistical models considered to estimate temporal trends in 100-year return levels of daily snowfall.

For small-size time series of annual maxima, e.g. few decades, return levels uncertainty largely depend on the uncertainty of the shape parameter (Koutsoyiannis, 2004). In order to reduce the uncertainty of the GEV parameters, models are often fitted using the information from several time series e.g. using regionalization methods (Hosking et al., 2009) or using scaling relationships between different aggregation durations (Blanchet et al., 2016). In this work, we fit models to time series from several elevations. In the literature, elevation is often not treated as a covariate, but rather accounted for by a spatial distance

(Blanchet and Davison, 2011; Gaume et al., 2013; Nicolet et al., 2016). In practice, as illustrated in Fig. 5, we compute the uncertainties of return levels for all the massifs and find that elevational-temporal models effectively decrease confidence intervals compared to pointwise distributions fitted to one time series.

    Then, we fit the models to at least two time series, i.e. at least from two elevations. As illustrated in Figure 1, for each range of elevations, we always have at least two time series, i.e. more than 100 maxima to estimate 100-year return levels. However,

in practice annual maxima from consecutive elevations are often dependent. At low elevations, four times series contain zeros, i.e. years without any snowfall, which may lead to misestimation for the models. A potential solution would be to rely on a mixed discrete–continuous distribution: a discrete distribution for the probability of a year without any snowfall, and a continuous GEV distribution for the annual maxima of snowfall. However, this would require at least one additional parameter for the discrete distribution, and even more if we wish to model some non-stationarity. In practice, to avoid overparameterized

models, we removed one time series which had more than 10% of zeros.

    Afterwards, for different ranges of elevation containing at most three consecutive elevations, we fit the models using all corresponding time series. For each model, this ensures that the temporal non-stationarity can be assumed to not depend on the elevation. Indeed, initially we intended for each massif to fit a single model to time series from all elevations. However, to account for decreasing trends at low elevations and increasing trends at high elevations, this leads to complex overparameterized

models that often did not fit well. We decided to consider a piecewise approach, i.e. simpler models fitted to ranges of consecutive elevations at most separated by 900 m (Fig. 1, Fig. 5). This ensures that we can assume that the temporal non-stationarity (no trend or decreasing/increasing trend) is shared between all elevations from the same range. Otherwise, we observe that annual maxima from consecutive altitudes are dependent (Fig. 1). We did not account for this dependence in our statistical model. Instead, we assumed that maxima are conditionally independent given the vector of parameters $\theta_{\mathcal{M}}$ (Sect. 3.2).

Finally, for each range of elevations, we consider models with a temporal non-stationarity only for the location and scale parameter. Indeed, in the literature, a linear non-stationarity is considered sometimes only for the location parameter (Fowler et al., 2010; Tramblay and Somot, 2018) but more often both for the location and the scale (or log-transformed scale for numerical reasons) parameters (Katz et al., 2002; Kharin and Zwiers, 2004; Marty and Blanchet, 2012; Wilcox et al., 2018). Otherwise, the shape parameter is typically considered temporally stationary in the literature, and we followed this approach.

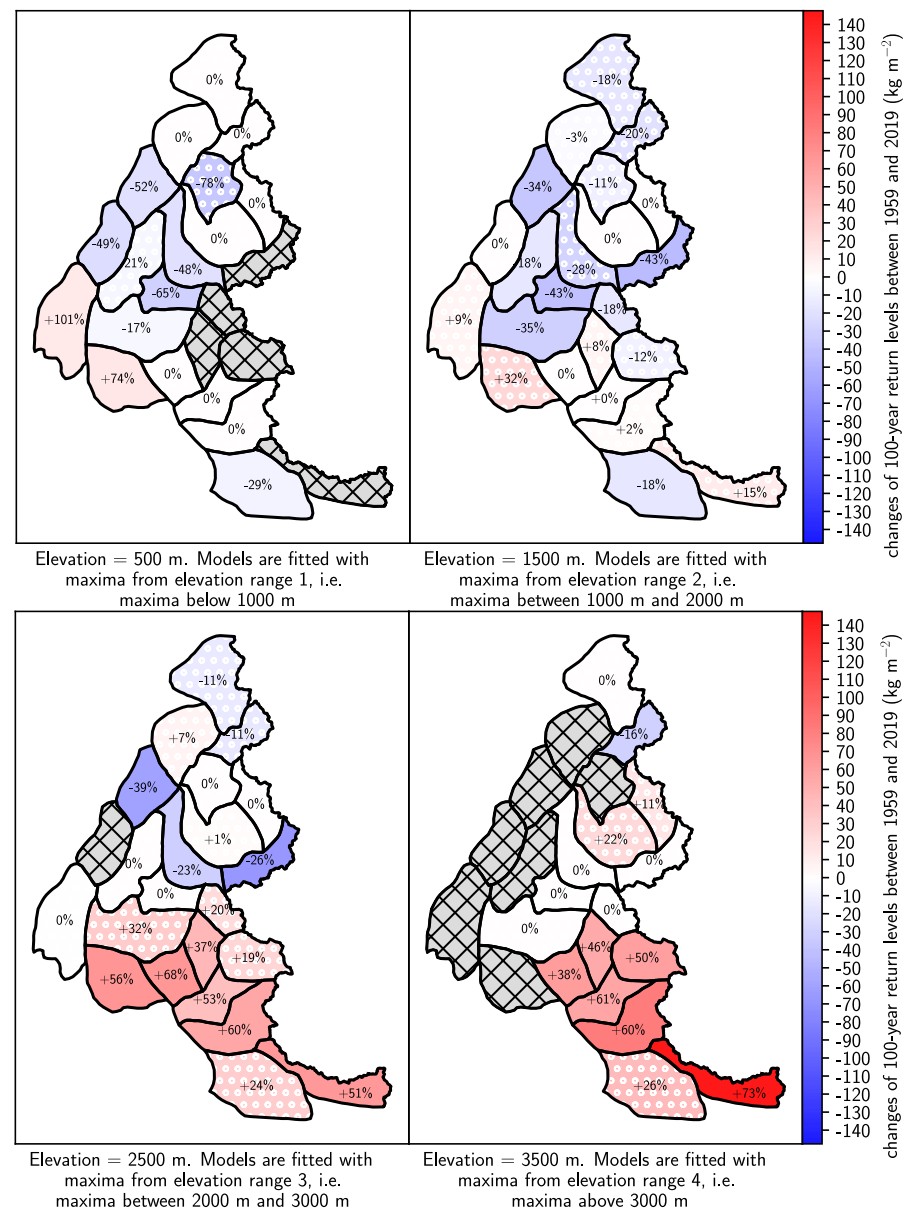

**Figure 8.** Changes of 100-year return levels of daily snowfall between 1959 and 2019 for each range of elevations. The corresponding relative changes are displayed on the map. Hatched grey areas denote missing data, e.g. when the elevation is above the top elevation of the massif. Changes of return levels are computed at the middle elevation for each range, e.g. at 1500 m for the range 1000-2000 m. Massifs with non-significant trends are indicated with a white-dotted pattern.

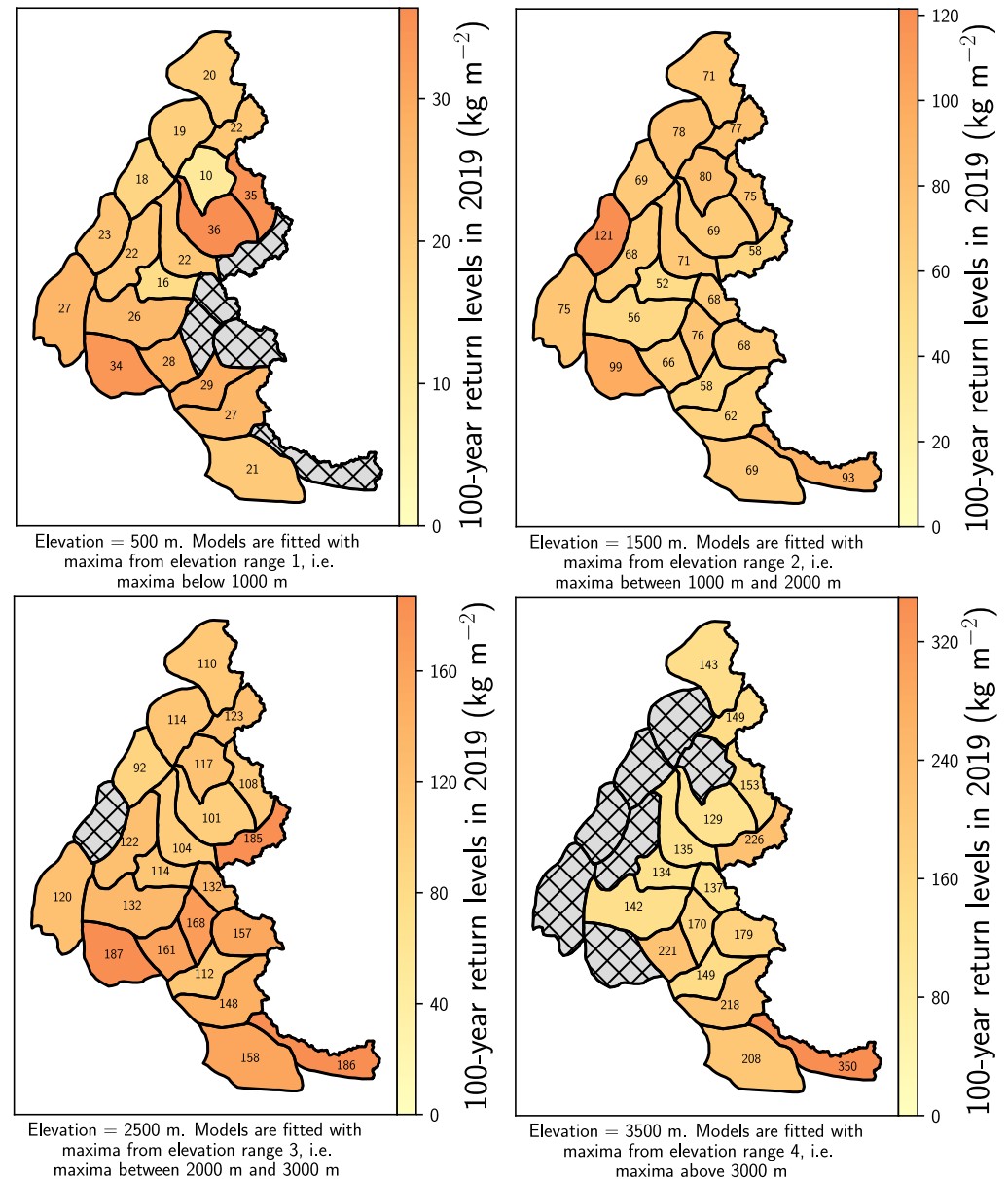

**Figure 9.** 100-year return levels in 2019 of daily snowfall for each range of elevations. 100-year return levels are both illustrated with colors (elevation range dependent scale) and written on the map. Hatched grey areas denote missing data, e.g. when the elevation is above the top elevation of the massif. Return levels are computed at the middle elevation for each range, e.g. at 1500 m for the range 1000-2000 m

## 5.2 Implication of the temporal trends in 100-year return levels

In Figure 8, we emphasize massifs with a strong increase of 100-year return levels between 1959 and 2019, e.g. massifs filled with medium/dark red correspond to increase $\geq +50$ kg m$^{-2}$. Settlements in these massifs should ensure that the design of protective measures and building standards against extreme snow events are still adequate after such an increase. Hopefully, this might concern a few settlements as such strong increases are always located above 2000 m. However, this might impact ski resorts which should ensure that the design of avalanche protections take this change into account. We note that extreme snow events can sometimes be triggered by one snowfall event, but often depend on other factors such as accumulated snow or wind. Therefore, to update structure standards for ground snow load (Biétry, 2005), we should account both for this increasing trend in annual maxima of daily snowfall, and trends in annual maxima of ground snow loads (Le Roux et al., 2020). Indeed, most known snow load destructions result from such intense and short snow events, sometimes combined with liquid precipitation, which is not considered in this study. In general, in mountainous regions around the French Alps, if the past trends continue into the future, for extreme snowfall we can expect decreasing trends below 1000 m, and increasing trends above 3000 m, as this agrees both with our results and the literature (Tab. 1).

## 5.3 Data considerations

Following the evaluation of the SAFRAN reanalysis cited in Sect. 2, we can conclude that this reanalysis most likely underestimates high-elevation precipitation (above 2000 m), which probably leads to an underestimation of high-elevation snowfall. This deficiency does not affect the main contribution of this article, i.e. a majority of decreasing trend below 2000 m, and of a majority of increasing trend above 2000 m. However, this deficiency affects the value of extreme snowfall, i.e. 100-year return level and the scale of their changes, which may be underestimated above 2000 m. For future works, we note that a direct evaluation of extreme snowfall could help to better pinpoint the locations where return levels might be biased.

## 5.4 Hypothesis for the contrasted pattern for changes of 100-year return levels

In Figure 8, at 2500 m, we find a spatially contrasted pattern for changes of 100-year return levels of snowfall: most decreasing trends are located in the north, while most increasing trends are in the south.

These changes contradict expectations based on the climatological differences between the north and south of the French Alps. Indeed, since the north is climatologically colder than the south both in winter and summer (Fig. 5 of Durand et al. 2009), we would have expected the inverse pattern for an intermediate elevation, i.e. increasing trends in the north, and decreasing trends in the south. Indeed, extreme snowfall stems from extreme precipitation occurring in a range of optimal temperatures slightly below $0^{o}$C (O'Gorman, 2014; Frei et al., 2018). Thus, under global warming, we would have expected for an intermediate elevation that the probability to be in the range of optimal temperatures would be increasing in the north (because mean temperatures would shift toward the optimal range) while decreasing in the south (because temperatures would be shifting away from the optimal range). Therefore, we would have observed an increase in extreme snowfall in the north, and a decrease in the south.

Thus, this spatial pattern of changes cannot be solely explained by the spatial pattern of mean temperature. In particular, we believe that dynamical changes, i.e. heterogeneous changes of extreme precipitation in the French Alps, may have contributed to generate this contrasted pattern. In Appendix D, we apply our methodology (Sect. 3) on daily winter precipitation from the SAFRAN reanalysis for the period 1959-2019. We focus on winter precipitation because winter is the season where most annual maxima of daily snowfall occur below 3000 m (Appendix E). We observe that at 2500 m (and at all elevations) that changes of 100-year return levels of winter precipitation show the same contrasted pattern than 100-year return levels of snowfall

This contrasted pattern is observed at all elevation for changes of 100-year return levels of winter precipitation. The literature also confirms that changes of extreme precipitation are not homogeneous. For instance, we observe for the period 1903-2010 that trends in daily maxima of winter precipitation are stronger in the south (+20-40% per century) compared to the north (from -10% to +20% per century) of the French Alps (Fig. 7 of Ménégoz et al. 2020). We also observe for the period 1958-2017 that 20-year return level of winter precipitation have decreased in the north of the French Alps and have slightly increased or remained the same in the south (Fig. 8 of Blanchet et al. 2021). This observation might be due to a stronger increasing trend in extreme precipitation for the Mediterranean circulation than for the Atlantic circulation. Indeed, precipitation maxima in the north of the French Alps are frequently triggered by the Atlantic circulation, while maxima in the south are often due to the Mediterranean circulation (Blanchet et al., 2020). Furthermore, increasing trends in extreme snowfall have already been observed at the proximity of the Mediterranean sea. For example, Faranda (2020) identified a certain number of Mediterranean countries showing positive changes of snowfall maxima. D'Errico et al. (2020) proposes a physical explanation of this phenomenon: the Mediterranean sea is warming faster than any other ocean, which enhances convective precipitation and favours heavy snowfalls during cold spell events.

In practice, this increasing trend in extreme snowfall in the south should be temporary. Indeed, with climate change, temperatures are expected to shift further away from the optimal range of temperatures for extreme snowfall. Thus, in the long run, extreme snowfall is expected to decrease as the increase of extreme precipitation shall not compensate for the decreasing probability to be close to the optimal range of temperatures.

## 6  Conclusions and outlooks

We estimate temporal trends in 100-year return levels of daily snowfall for several ranges of elevation based on the SAFRAN reanalysis available from 1959 to 2019 (Durand et al., 2009). Our statistical methodology relies on non-stationary extreme value models that depend both on elevation and time. Our results show that a majority of trends are decreasing below 2000 m and increasing above 2000 m. Quantitatively, we find an increase of 100-year return levels between 1959 and 2019 equal to +23% (+32 kg m$^{-2}$) on average at 3500 m, and a decrease of -10% ($-7$ kg m$^{-2}$) on average at 500 m. For the four investigated elevation ranges, we find both decreasing and increasing trends depending on location. In particular, we observe a spatially contrasted pattern, exemplified at 2500 m: 100-year return levels have decreased in the north of the French Alps while increased in the south. In the discussion, we highlight that this pattern might be related to known increasing trends in extreme snowfall at the proximity of the Mediterranean Sea.

Many potential extensions of this work could be considered. First, reanalyses are increasingly available at the European scale (e.g. Soci et al. 2016), which could be used for extending this work to a wider geographical scale. In this case, instead of considering close massifs as spatially independent, we believe that our methodology may benefit from an explicit modelling of the spatial dependence (Padoan et al., 2010). Then, climatic projections could enable us to explore temporal trends up to the end of the twenty-first century. In these circumstances, it might be more relevant to use global mean surface temperature

as a temporal covariate to combine ensembles of climate models. Finally, future research should not focus solely on mountain regions but also on lowland regions such as around the Mediterranean sea. Indeed, such regions are often heavily impacted by snow related hazards because they are ill-equipped for such rare events. For instance, extreme snowfall over Roussillon, a Mediterranean coastal lowland, caused major damages in 1986 (Vigneau, 1987), while in 2021 heavy snowfall over Spain caused at least 1.4 billion Euro of damage (The New York Times, 2021). In these regions, temperatures below the rain-snow

transition temperature, i.e. roughly below $0^o$C, may tend to be rare in the future. Therefore, in these cases, in addition to directly studying trends in snowfall extremes, we should focus on trends in the compound risk of cold-wet events (De Luca et al., 2020).

## Appendix A:  Quantile-quantile plots

A quantile-quantile (Q-Q) plot is a standard diagnosis tools based on the comparison of empirical quantiles (computed from the

empirical distribution) and theoretical quantiles (computed from the expected distribution). For non-stationary extreme value models, the approach is two-fold (Coles, 2001; Katz, 2012). First, we transform observations into residuals with a probability integral transformation $f_{\text{GEV}\rightarrow\text{Standard Gumbel}}$. Then, we construct a Q-Q plot to assess if the residuals follow a standard Gumbel distribution. If the Q-Q plot reveals a good fit it means that the non-stationary extreme value model has a good fit as well.

We start by transforming the observations $y_{z_1,t_1},...,y_{z_1,t_M},...,y_{z_N,t_1}...,y_{z_N,t_M}$ into residuals. Let $Y_{z,t} \sim \text{GEV}(\mu(z,t),\sigma(z,t),\xi(z))$

with parameter $\boldsymbol{\theta}_{\mathcal{M}}$. By definition of the probability integral transformation $f_{\text{GEV}\rightarrow\text{Standard Gumbel}}$, we obtain that:

$$f_{\text{GEV}\rightarrow\text{Standard Gumbel}}(Y_{z,t};\boldsymbol{\theta}_{\mathcal{M}}) = \frac{1}{\xi(z)}\log(1+\xi(z)\frac{Y_{z,t}-\mu(z,t)}{\sigma(z,t)}) \sim \text{Gumbel}(0,1). \tag{A1}$$

The transformed observations, a.k.a. residuals, are denoted as $\epsilon_{z,t} = f_{\text{GEV}\rightarrow\text{Standard Gumbel}}(y_{z,t};\boldsymbol{\theta}_{\mathcal{M}})$. Afterwards, we construct a Q-Q plot to assess if the residuals follow a standard Gumbel distribution. On one hand, the $N \times M$ empirical quantiles correspond to the ordered values of the residuals $\epsilon_{z_1,t_1},...,\epsilon_{z_1,t_M},...,\epsilon_{z_N,t_1}...,\epsilon_{z_N,t_M}$. On the other hand, we compute

the corresponding $N \times M$ theoretical quantiles, which are the quantile $\frac{i}{N\times M+1}$ of the standard Gumbel distribution, where $i \in \{1,...,N \times M\}$.

In Figure A1, we display Q-Q plots for the selected model for the time series displayed in Figure 1. We observe that they show a good fit, as the points stay close to the line. In general, most retained models show a good fit. Furthermore quantitatively, if we rely on an Anderson–Darling statistical test with 5% significance level to assess if the residuals follow a standard Gumbel

distribution, we find that the largest part of tests are not rejected (not shown).

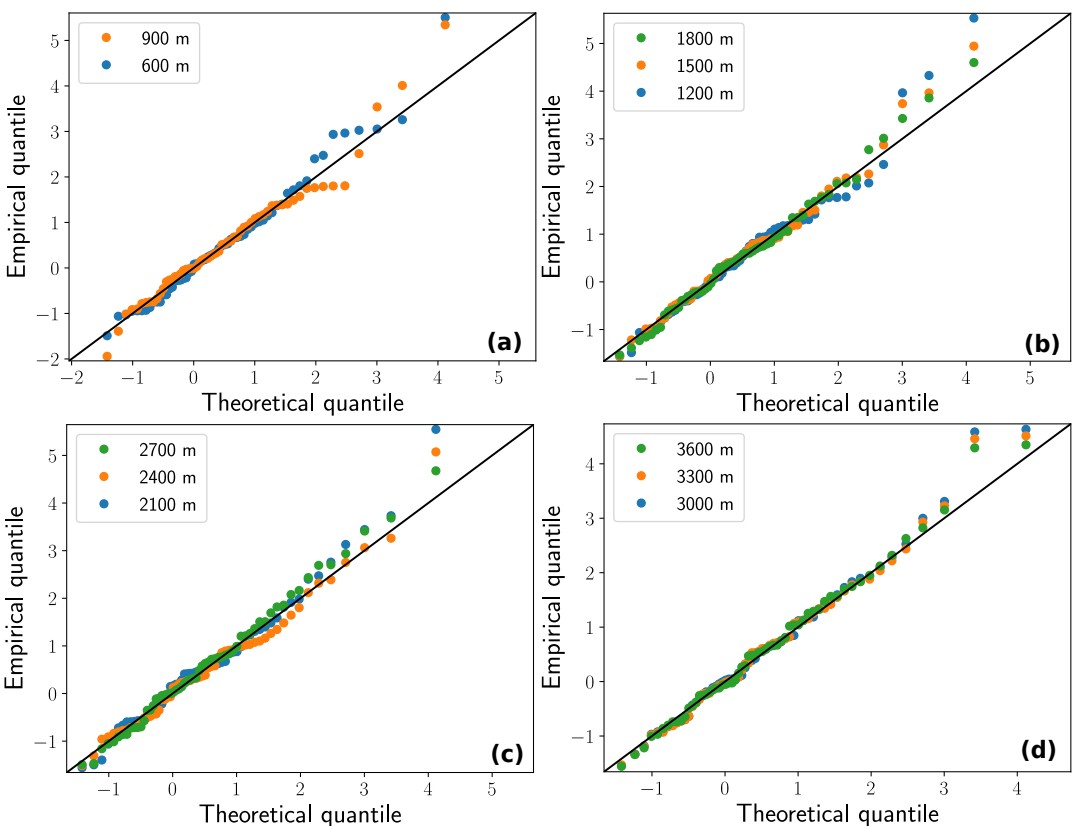

**Figure A1.** Q-Q plots of the selected elevational-temporal models for the Vanoise massif for the four ranges of elevations considered (see Fig. 1 for the time series). **(a)** Below 1000 m. **(b)** Between 1000 m and 2000 m. **(c)** Between 2000 m and 3000 m. **(d)** Above 3000 m.

## Appendix B: Semi-parametric bootstrap method

In the context of maximum likelihood estimation, uncertainty related to return levels can be evaluated with the delta method, which quickly provides confidence intervals both in the stationary and non-stationary cases (Coles, 2001; Gilleland and Katz, 2016). However, due to the dependence between maxima from consecutive elevations (Fig. 1), we decided to compute confidence intervals with a bootstrap resampling method (Efron and Tibshirani, 1993). This resampling method allows us to estimate the uncertainties resulting from in-sample variability. In this article, we rely on a semi-parametric bootstrap resampling method adapted to non-stationary extreme models (Kharin and Zwiers, 2004; Sillmann et al., 2011).

We generate $B = 1000$ bootstrap samples using the parameter $\widehat{\boldsymbol{\theta}}_{\mathcal{M}}$. For each bootstrap sample $i$, the semi-parametric bootstrap method is four-fold. First, as explained in Appendix A, we compute the residuals $\epsilon_{z_1,t_1}, ..., \epsilon_{z_1,t_M}, ..., \epsilon_{z_N,t_1} ..., \epsilon_{z_N,t_M}$. Then, from these residuals we draw with replacement a sample of size $M \times N$. We denote these bootstrapped residuals as $\tilde{\epsilon}_{z_1,t_1}, ..., \tilde{\epsilon}_{z_1,t_M}, ..., \tilde{\epsilon}_{z_N,t_1} ..., \tilde{\epsilon}_{z_N,t_M}$. Afterwards, we transform these bootstrapped residuals into bootstrapped annual maxima

as follows: $\tilde{y}_{z,t} = f^{-1}_{\text{GEV} \to \text{Standard Gumbel}}(\tilde{\epsilon}_{z,t}; \boldsymbol{\theta}_{\mathcal{M}})$. Finally, we estimate the parameter $\widehat{\boldsymbol{\theta}}^{(i)}_{\mathcal{M}}$ of model $\mathcal{M}$ with the bootstrapped annual maxima $\tilde{y}_{z_1,t_1}, ..., \tilde{y}_{z_1,t_M}, ..., \tilde{y}_{z_N,t_1} ..., \tilde{y}_{z_N,t_M}$. To sum up, this bootstrap procedure provides a set $\{\widehat{\boldsymbol{\theta}}^{(1)}_{\mathcal{M}}, ..., \widehat{\boldsymbol{\theta}}^{(i)}_{\mathcal{M}}, ..., \widehat{\boldsymbol{\theta}}^{(B)}_{\mathcal{M}}\}$ of $B$ parameters for the model $\mathcal{M}$.

In practice, we rely on this set of GEV parameters to obtain $80\%$ confidence intervals for 100-year return levels (Fig. 5), or for time derivative of 100-year return levels (Sect. 3.3). For instance, in the latter case we have that $\hat{p}(\frac{\partial y_{0.01}(z,t)}{\partial t} > 0 | \widehat{\boldsymbol{\theta}}_{\mathcal{M}}) = \frac{1}{B}\sum_{i=1}^{B} \mathbb{1}_{\frac{\partial y_{0.01}(z,t)}{\partial t}^{(i)} > 0}$, where $\frac{\partial y_{0.01}(z,t)}{\partial t}^{(i)}$ is the time derivative of 100-year return levels for the parameter $\widehat{\boldsymbol{\theta}}^{(i)}_{\mathcal{M}}$.

## Appendix C:  Sensivity to the return period

The 100-year return period was chosen because it is the largest return period considered inthe Eurocode to build structures
(Cabrera et al., 2012). We believe that this return period is the most familiar return period for non-experts as it corresponds to a centennial event. For smaller return periods (5-10 years), our results also apply. In Figure C1, we illustrate our results for the 10, 50 and 100-year return periods. We observe that the overall distribution of increasing/decreasing trend for the return levels is almost insensitive to the choice of the return period. For instance, the only noticeable difference between the 10 and 100 return periods, is that for the elevation range 1000-2000m and for the elevation range 2000-3000 m, we observe that one
massif shows an increasing trend for the return period 10 years, while it is decreasing for the return period 100 years.

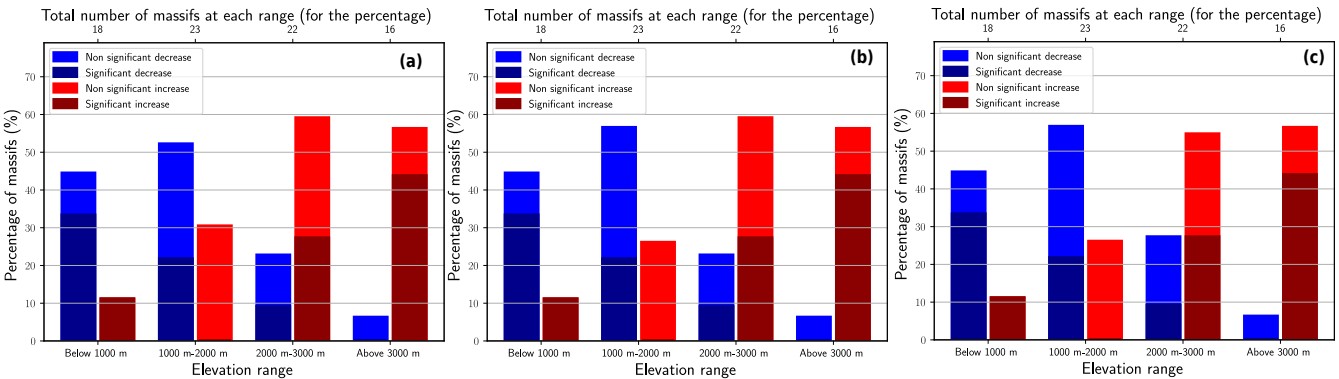

**Figure C1.** Percentages of massifs with significant/non-significant trends return levels of daily snowfall for each range of elevation and for a return period equals to **(a)** 10, **(b)** 50 or **(c)** 100 years. A massif has an increasing/decreasing trend if the return level of the selected elevational-temporal model has increased/decreased.

## Appendix D:  Trends in 100-year return levels of winter precipitation

We apply the same methodology as our study (Sect. 3) to daily winter (December to February) precipitation obtained with the SAFRAN reanalysis, and spanning the period 1959-2019. First, a preliminary analysis with pointwise fits indicates that a linear parametrization w.r.t. the elevation for the location and scale parameters is also valid for the winter precipitation (Fig. D1). In

Figure D2, we illustrate changes of 100-year return level of winter precipitation. We observe a spatially contrasted pattern at all elevations, i.e. increase in the south, and decrease in the north. This underlines that the spatially contrasted pattern observed for changes of 100-year return level of snowfall at 2500 m may result from the circulation patterns of precipitation.

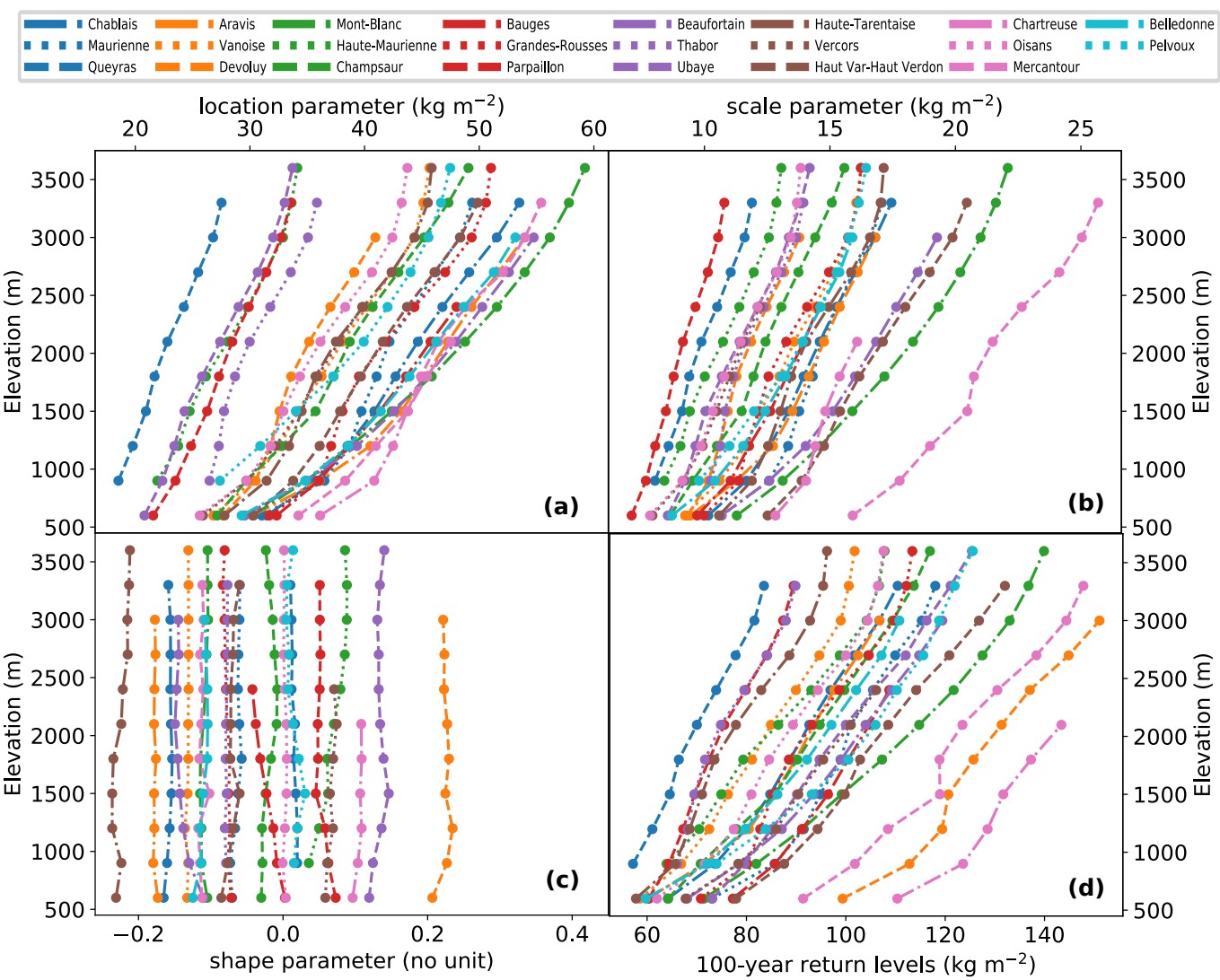

**Figure D1.** Changes of GEV parameters **(a, b, c)** and of 100-year return levels **(d)** with the elevation for the 23 massifs of the French Alps. GEV distributions are estimated pointwise for the annual maxima of daily winter precipitation every 300 m of elevation.

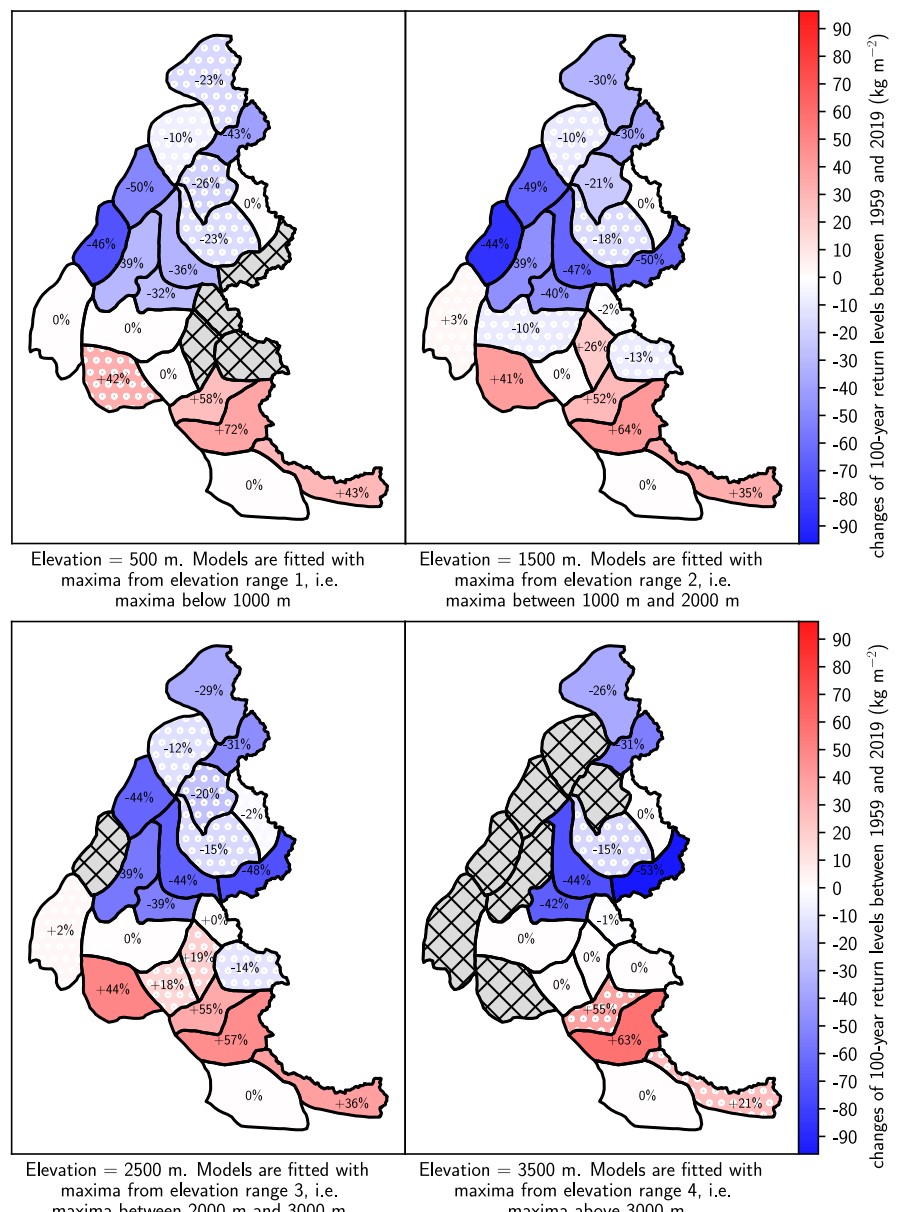

**Figure D2.** Changes of 100-year return levels of daily winter precipitation between 1959 and 2019 for each range of elevations. The corresponding relative changes are displayed on the map. Hatched grey areas denote missing data, e.g. when the elevation is above the top elevation of the massif. Changes of return levels are computed at the middle elevation for each range, e.g. at 1500 m for the range 1000-2000 m. Massifs with non-significant trends are indicated with a white-dotted pattern.

## Appendix E:  Seasons when the annual maxima of daily snowfall occurred

In Figure E1, we study the seasons when the annual maxima of daily snowfall occurred. For the elevation range 1 (below 1000

385   m) and for the elevation range 2 (between 1000 m and 2000 m), we observe that the annual maxima mainly occurred (>60%) between December and February, i.e. the coldest part of the snow season. For the elevation range 3 (between 2000 m and 3000 m) more than 40% of maxima occurred in winter, while slightly less than 30% occurred in autumn and in spring. For the elevation range 4 (above 3000 m) the seasons of occurrence are more spread, even if we observe that more than 40% of maxima occurred in autumn. In conclusion, we find that below 3000 m, most annual maxima of daily snowfall occur in winter,

while above 3000 m they mostly occur in autumn..

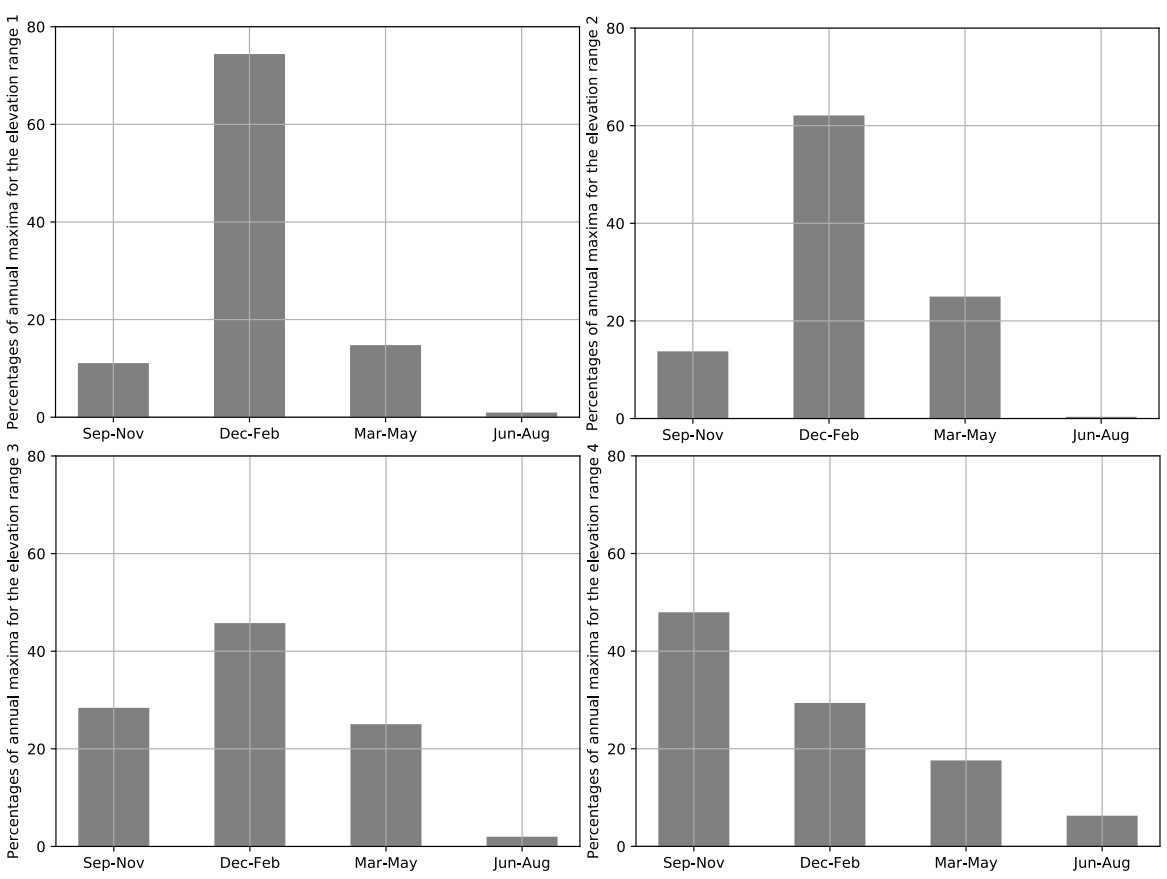

**Figure E1.** Seasons when the annual maxima of daily snowfall occurred for the elevation range 1 (below 1000 m), the elevation range 2 (between 1000 m and 2000 m), the elevation range 3 (between 2000 m and 3000 m) and the elevation range 4 (above 3000 m).

*Author contributions.* ELR, GE and NE designed the research. ELR performed the analysis and drafted the first version of the manuscript. All authors discussed the results and edited the manuscript.

*Competing interests.* The authors declare that they have no conflict of interest.

*Data availability.* The full SAFRAN reanalysis on which this study grounds is freely avalaible on AERIS (Vernay et al., 2019).

*Acknowledgements.* E. L. R. holds a PhD grant from INRAE. We are grateful to Eric Gilleland for his "extRemes" R package, and Ali Saeb for his "gnFit" R package. Inrae, CNRM and IGE are members of Labex OSUG.

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
