# Peer review of "Elevation-dependent trends in extreme snowfall in the French Alps from 1959 to 2019"

_The Cryosphere, 2021_

## Author Comment (AC1)

We thank Reviewer #1 for his/her useful questions and comments on our manuscript. Please find below a detailed feedback to individual comments and questions.

Main comment:

The single main deficiency of the work is the missing validation of the SAFRAN data with respect to extreme snowfall amounts and their temporal trends. Several previous works are cited, but much more information is necessary in my opinion. Potential deficiencies of the reanalysis dataset in representing extreme snowfall woudl ultimately affect the conclusions of this work. I strongly suggest that the authors try to better motivate the use of this reanalysis dataset for their purpose.

**Motivation for the SAFRAN reanalysis.** The main motivation to rely on the SAFRAN reanalysis is that it provides consistent data to study the evolution of meteorological conditions and their impacts. Indeed, the direct use of observations have several drawbacks: they are intrinsically uncertain, like any observation, they often contain temporal gaps, and pose, like any point observation, representativeness challenges due to local characteristics of the location of the observations points (orography, vegetation). On the other hand, the SAFRAN reanalysis provides "augmented observations" that draw on all available observations and are supplemented with physical laws driving the numerical weather prediction model used for the reanalysis process (mass, momentum and energy conservation within the meteorological), and can reduce uncertainty by gathering strength from the combination of multiples sources of observations (temperature, precipitation). Indeed, the SAFRAN reanalysis includes precipitation in-situ data in its analysis scheme (in contrast to other popular global reanalysis products such as ERA-Interim/ERA5 reanalyses).

We further consider that we cannot "validate" SAFRAN but rather "evaluate" it, like for any geophysical product. Indeed, validating SAFRAN would imply that in situ measurements should be considered as a reference because they directly measure snowfall, and are therefore considered as the truth. However, all observations carry uncertainty and this is particularly the case for in situ snowfall measurements (see e.g. Nitu et al., 2018). Firstly, they cannot span the spatial heterogeneity, which is critical in mountainous areas (local effects due to the topography, winds, etc.). Snowfall measurements are also very uncertain for high precipitation intensities, usually associated with strong winds due to gauge undercatch issues. They are also not necessarily consistent due to the variety of measurement tools and types used in the measurement networks (automated, manual, temporal resolution).

**Evaluation of the SAFRAN reanalysis.** A specific evaluation of SAFRAN for extreme snowfall goes beyond the scope of this article and could easily result in a scientific publication on its own. In our study, we already mention that the "SAFRAN reanalysis has been evaluated both directly with in situ temperature and precipitation observations and in-directly with various snow depth observations". In the revised version of the manuscript, we will add the following additional comparisons:

- *"In Vionnet et al. (2019), the SAFRAN reanalysis has been evaluated for snowfall against two numerical weather prediction (NWP) systems for the winter 2011-2012. They find that seasonal snowfall averaged over all the massifs of the French Alps reaches 546 mm in SAFRAN, 684 mm in the first NWP, and 737 mm in the second NWP. In details, they find that SAFRAN significantly differ from the two NWP systems in (i) areas of high elevation, probably due to the limited number of high-elevation stations and gauge undercatch (ii) on the windward side of the different mountain*

*ranges due to the assumption of climatological homogeneity within each SAFRAN massif."*

● *"In Menegoz et al. (2020), the SAFRAN reanalysis has been compared to the regional climate model MAR which uses ERA-20C as forcing. They found that the vertical gradient of annual mean of total precipitation of SAFRAN is generally smaller than those simulated by MAR."*

Therefore, the SAFRAN reanalysis most likely underestimates high-elevation precipitation (above 2000 m), which probably leads to an underestimation of high-elevation snowfall. This deficiency does not affect the main conclusion of our article, with a majority of decreasing trend below 2000 m, and of a majority of increasing trend above 2000 m. However, this deficiency affects the value of extreme snowfall, i.e. 100-year return level (Fig. 9) and the scale of changes of 100-year return level (Fig. 8), which may be underestimated above 2000 m.

Minor comments:

Line 78-80: This section is far too short, a much better motivation for using SAFRAN instead of, for instance, station series, needs to be provided (see above). A dedicated validation exercise would help.

For an answer to this suggestion, we refer to our answer to the Main comment.

Chapter 4.1 and further: I'm not sure if the term "pointwise" should be used here. I understand the meaning and the difference to "piecewise" in statistical terms, but pointwise could be misunderstood as being based on observations taken at individual locations/points (which is not the case here)

We understand that these terms could be misunderstood. Thus, in the revised manuscript, we will add the two following sentences in In Sect. 3.4 to clarify this fact.

● *"Pointwise distribution stands for a distribution fitted on the annual maxima from a single elevation of some massif."*

● *"Piecewise models stands for models fitted on the annual maxima from a range of elevation of some massif."*

Lines 189-199: This section is basically a very brief description of one figure after the other. It should be extended and some more information on each figure and a brief description of what they show and what this means) needs to be provided.

We agree that the description for Figure 8 is rather short. In the revised manuscript, we will add the following informations for the changes of 100-year return levels:

*"In Figure 8, we display the change of 100-year return levels between 1959 and 2019 for each range of elevations. At 500 m, we observe that eight massifs have a stationary trend, and five massifs located in the center of the French Alps have a significant decreasing trend. We also note that two massifs located in the Western French Alps have an increasing significant trend, with an absolute change of 100-year return level close to +20 kg m$^{-2}$. At 1500 m, six massifs in the center of the French Alps have decreasing trends. At 2500 m, we observe a spatially contrasted pattern: most decreasing trends are located in the north, while*

*most increasing trends are located in the south. We discuss this pattern in Sect. 5.4. At 3000 m, we observe that the six massifs with a significant increasing trend are located in the South of the French Alps.*"

Chapter 5.1: This chapter is very important as it highlights potential limitations of the work. However, the implications of these limitations for the interpretation of the results and for the conclusions remain largely unclear. Are the conclusions valid nevertheless or do they have to be questioned?

In this section 5.1 we highlight the main hypothesis of our study:

- annual maxima from a range of elevation of some massif are assumed to have the same temporal trend,
- conditional independence of the annual maxima given the vector of parameters,
- the shape parameter is considered constant.

We do not believe that these hypotheses are limitations, but rather "classical" assumptions that increase the robustness of our conclusions by avoiding, e.g., overparameterization. Firstly, we believe that the former hypothesis is reasonable. Then, the two latter hypotheses are standard in the field of extreme value analysis, and seem a good choice given the fact that we only have 61 annual maxima for each elevation. Indeed, instead of the "conditional independence" hypothesis, an explicit modeling of dependence (with max-stable processes for instance) would lead to over-parametrized models. Moreover, a time-dependent shape parameter can distort the form of the distribution w.r.t. years, which is likely to lead to overfitting when we model changes w.r.t. 61 years.

Lines 244-245: This implication does actually only hold if the past trends would continue into the future. Do you have any evidence for this?

We will replace the potentially  misleading sentence "*should ensure that protective measures can cope with this increase.*" with "*should ensure that protective measures are still valid after such an increase*."

Lines 275-277: What this basically means is that mean temperature is not the only control. This could be written much clearer.

We will add an introductory sentence to this paragraph: "*Thus, this spatial repartition of changes cannot be solely explained with the spatial repartition of mean temperature*."

**References**

Ménégoz, M., Valla, E., Jourdain, N., Blanchet, J., Beaumet, J., Wilhelm, B., … Anquetin, S. (2020). Contrasting seasonal changes in total and intense precipitation in the European Alps from 1903 to 2010. *Hydrology and Earth System Sciences*, 1–37. Retrieved from https://doi.org/10.5194/hess-24-5355-2020

Nitu, R., Roulet, Y. A., Wolff, M., Earle, M., Reverdin, A., Smith, C., ... & Yamashita, K. (2018). *WMO Solid Precipitation Intercomparison Experiment (SPICE)*. Tech. Rep., World Meteorological Organization, 2018. a.
https://library.wmo.int/doc_num.php?explnum_id=5686

Vionnet, V., Six, D., Auger, L., Dumont, M., Lafaysse, M., Quéno, L., … Vincent, C. (2019). Sub-kilometer Precipitation Datasets for Snowpack and Glacier Modeling in Alpine Terrain. *Frontiers in Earth Science*, 7(August), 1–21. https://doi.org/10.3389/feart.2019.00182

---

## Author Comment (AC2)

**Suggestions from Referee #2**

We thank Reviewer #2 for his/her useful questions and comments on our manuscript. Please find below detailed feedback to individual comments and questions.

Minor comments:

 -The first comment is about the selection of the 100 yr return period, as targeted intensity of heavy snow event in this work. Perhaps authors should justify more why this recurrence (that involves very rare events) was chosen. I think that replicate some of the analysis (i.e. Figure 6) for a more frequent recurrent time (i.e. 5-10 yr) could give good information about the sensitivity of the results to the return period selection, and in case there are significant diferences the results of the paper could be more interesting for a management point of view. At least, I think this question should be discussed.

The 100-year return period was chosen because it is the largest return period considered in the Eurocode to build structures (Cabrera et al. 2021). We believe that this return period is the most familiar return period for non-experts as it corresponds to a centennial event. For smaller return periods (5-10 years), our results also apply. The Figure below provides results with a return period of 10 years and with a return period of 100 years, the overall distribution of increasing/decreasing trend for the 10-year return period remains the same as for the 100-year return period. The only noticeable difference is that for the elevation range 1000-2000m and for the elevation range 2000-3000 m, we observe that 1 massif shows an increasing trend for the return period 10 years, while it is decreasing for the return period 100 years.

[Figure]

Legend: Percentages of massifs with significant/non-significant trends in **10-year (Left) 100-year (Right) return levels of daily snowfall** for each range of elevation.A massif has an increasing/decreasing trend if the 10-year return level of the selected elevational-temporal model has increased/decreased.

We will add the following sentence in the manuscript: "*We note that the sign and the significance of the trends (summarized with the percentages on Fig. 6) remain more or less similar for the trends in 10-year and 50-year return periods events.*" In an appendix section, we will add the equivalent of Figure 6 and Figure 8 for the 10-year and 50-year return level.

- Another question is that the assignation of existing trends to warmer climate or changes in precipitation intensity is discussed in a very qualitative way (based on some references) when this is a very interesting topic from a climate change perspective, as associated

uncertainties of precipitation intensities are much larger than the ones for temperature warming. Perhaps simply presenting a map of temperature and 100 yr precipitation intensity during the snow season, and may be simple cross tabulation test could answer much of this question.

The works cited about changes in extreme precipitation over the Alps do not present in most cases their respectives study periods. They should be presented as trends on this parameter may change a lot depending on the selected period, and only those covering a similar time span than this study can be used as reference.

We understand that the main issue is that the works cited on changes in extreme precipitation over the Alps do not correspond to our study period, i.e. 1959-2019. Therefore, following the advice of reviewer #2, we present (see below) a map of temperature and 100-year precipitation intensity during the winter (December to February) obtained with the SAFRAN reanalysis, and spanning the period 1959-2019. For the temperature, we directly compute the mean winter temperature. For the changes of 100-year return level of winter precipitation, we follow the same methodology as our study.

[Figure]

Legend: Mean winter temperature averaged for the period 1959-2019 for each range of elevations. The mean temperature is written on the map. Hatched grey areas denote missing data, for example when one of the elevations in the range is above the top elevation of the massif.

The temperature maps clearly show that in the French Alps "the north is climatologically colder than the south" for the 4 elevation ranges.

Then, for the 100-year precipitation intensity, we applied the same methodology as our study. Indeed, a preliminary analysis with pointwise fits indicates a linear parametrization w.r.t. the elevation for the location and scale parameters.

[Figure]

Legend: Changes of GEV parameters (a,b,c) and of 100-year return levels (d) with the elevation for the 23 massifs of the French Alps. GEV distributions are estimated pointwise for the **annual maxima of winter precipitation** every 300 m of elevation.

We observe a contrasted pattern for the total precipitation at all elevations. Above 2000 m, we observe that maxima often occurs in autumn (see answer to the next suggestion on the seasonality). We applied the same methodology on the total precipitation in autumn, and found the same contrasted pattern (not shown).

[Figure]

Elevation = 500 m. Models are fitted with maxima from elevation range 1, i.e. maxima below 1000 m

Elevation = 1500 m. Models are fitted with maxima from elevation range 2, i.e. maxima between 1000 m and 2000 m

Elevation = 2500 m. Models are fitted with maxima from elevation range 3, i.e. maxima between 2000 m and 3000 m

Elevation = 3500 m. Models are fitted with maxima from elevation range 4, i.e. maxima above 3000 m

Legend: Changes of 100-year return levels of daily **winter precipitation** between 1959 and 2019 for each range of elevations. The corresponding relative changes are displayed on the map. Hatched grey areas denote missing data, e.g. when the elevation is above the top elevation of the massif. Changes of return levels are computed at the middle elevation for each range, e.g. at 1500 m for the range 1000-2000 m. Massifs with non-significant trend are indicated with a white-dotted pattern

To conclude, these analyses underline that the contrasted pattern of trends in 100-year return level of snowfall may result from the circulation patterns. In the revised manuscript, we will add an appendix section containing the two last figures, which justify the use of the same methodology to compute trends in 100-year return level of winter precipitation, and highlight at all elevations the contrasted pattern for trends in 100-year return level of winter precipitation.

- It would be also good to have an idea in which period of the snow season happen more frequently the very intense heavy snowfall events. The sensitivity of these events to climate change is supposed to be very different if they tend to happen in the coldest part of the snow season, or during the shoulder periods. This could be also a explanatory factor the the spatial heteregoneity shown between massifs.

In the following plots, we study the months when the annual maxima of snowfall occurred. We chose not to include this analysis in the revised manuscript, because we believe that it goes beyond the scope of this article.

For the range 1 (below 1000 m) and for the range 2 (between 1000 m and 2000 m), we observe that the annual maxima are mainly located (>60%) between December and February, i.e. the coldest part of the snow season.

[Figure]

Legend: Distribution of the month when the annual maxima of daily snowfall occurred for the period 1959-2019 for the range 1 (Left) and the range 2 (Right).

For the range 3 (between 2000 m and 3000 m) and for the range 4 (above 3000 m) the months of maxima are more spread. For the range 3, maxima occurred between November and March (each month has at least 10% of the maxima). For the range 4, maxima occurred between September and December. In both cases, we note that most of the distribution of maxima is centered on the early winter period (November-December).

We note that for all ranges of elevation we do not find a large difference in terms of seasonality between the massifs in the north of the French Alps, and the massifs in the South of the French Alps.

[Figure]

Legend: Distribution of the month when the annual maxima of daily snowfall occurred for the period 1959-2019 for the range 3 (Left) and the range 4 (Right).

In conclusion, we find that below 2000 m, annual maxima of daily snowfall mainly occurs between December and February, while above 2000 m it mainly occurs between November and December.

- In line 25, and later in discussion, is mentioned that optimal temperature for heavysnowfall events is slightly below 0oC. Probably in Alps may be tru, associated to the humid arrival of oceanic/mediterranean air masses, but this is not generalizable worldwide (I guess that heavy sowfalls in Colorado or Hokkaid will happen well below 0oC). I would just clarify the sentence.

This optimal temperature interval for extreme snowfall is not specific for the French Alps but stems from physics theory provided by O'Gorman (2014). In the revised manuscript, we will reformulate the sentence as follows: *"Extreme snowfall stems from extreme precipitation occurring in a range of optimal temperatures slightly below 0ºC according to physics theory (O'Gorman, 2014)."*

- I agree with the other review that a stronger validation of the dataset would be desiderable, but I also wonder how to make it properly, as for my knowledge the best observations of snow in the region have been use to be assimilated in the SAFRAN-CROCUS. This difficults a comparison with observations. May be this question should be mentioned in methods or discussion section

We agree that this is a critical point which needs to be further discussed (see our response to the reviewer #1).

**References**

Cabrera, A. T., Heras, M. De, Cabrera, C., & Heras, A. M. De. (2012). the Time Variable in the Calculation of Building Structures . How, 1–6. Retrieved from http://oa.upm.es/22914/1/INVE_MEM_2012_152534.pdf

O'Gorman, P. A. (2014). Contrasting responses of mean and extreme snowfall to climate change. *Nature*, *512*(7515), 416–418. https://doi.org/10.1038/nature13625

---

## Author Comment (AC3)

We thank Reviewer #3 for his/her useful questions and comments on our manuscript. Please find below detailed feedback to individual comments and questions.

Major comments:

Extreme snowfall or even snowfall in general from Safran reanalysis, at least to my knowledge, has so far never been evaluated with measurements as e.g. described in Gaume et al. (2013) or Nicolet et al. (2016). I believe it's necessary to show at least some comparisons with the available time series.

We agree that this is a critical point which needs to be further discussed (see our response to the reviewer #1).

Moreover, the following points are missing, which are both important in case of comparisons with measured snowfall water equivalents: a) Is the reanalysis able to reproduce rain-on-snow-events within a 24 h time period? b) Are the reanalysis data refer to the same 24h period as the measurements?

The SAFRAN reanalysis consists of hourly rainfall and snowfall values, so that when it is used as input to a snow cover model (e.g. the Crocus snow cover model as part of the S2M reanalysis), rain-on-snow events can be specifically analyzed. However, in the case of this particular study, we only focus on snowfall (precipitation) data hence we do not specifically analyze rain-on-snow events. The SAFRAN reanalysis system primarily operates at the daily time scale, using precipitation data from 6:00 to 6:00, which are then disaggregated to hourly time resolution as a separate part of the SAFRAN system. The daily data used here in the analysis are also integrated from 6:00 to 6:00, which minimizes any influence of the time disaggregation part of SAFRAN on daily precipitation values.

The examples of extreme snowfall or roof collapses provided are used to justify the use of the 100-year level, although all these examples are not caused by one extreme snowfall alone. Most of the examples also concern elevations below 600 m, which are not analyzed in this study at all. Please elaborate.

The 100-year return period was chosen because it is the largest return period considered in the Eurocode to build structures (Cabrera et al. 2021). We also believe that the 100-year return period is the most familiar return period for non-experts as it corresponds to a centennial event. Following the suggestion of referee #2, we will add in an Appendix, the equivalent of Figure 6 and Figure 8 for the trend in 10-year return and the 50-year return level.

We agree with Reviewer #3 that roof collapses are often not caused by one extreme snowfall event alone. As mentioned in another answer to one the suggestions of Reviewer #3, we will add the sentence "*extreme snow events are often triggered by one snowfall event but depend on other factors such as accumulated snow or wind.*"

The examples have been moved in the Perspective section of the article, where we argue that specific statistical tools are needed to study extreme snowfall in low-land areas below 600 m such as compound risk analysis, or mixed discrete-continuous distributions to handle the high number of annual maxima equal to zeros, which correspond to years without any snowfall.

The linear elevation dependence considered in L100 is in contradiction to doi.org/10.1029/2009WR007916. Please elaborate.

Indeed in paragraph 23 of Blanchet et al. (2009), there is a detailed analysis about the change in the location parameter with elevation. They find that this parameter strongly increases below 1200 m, and less strongly between 1200 m and 2200 m. This agrees with our piecewise linear elevation dependence. However, they also find that above 2200 m, the location parameter is almost constant with the elevation. Indeed, this is different with the linear dependence that we find for all the massifs (including the massifs close to the Swiss border) in our preliminary analysis (Fig. 2).

We note that the word "contradiction" seems strong, because it can be seen in Figure 3 of Blanchet et al. 2009 that the location parameter still increases slightly above 2200 m. This "contradiction" might be due to various differences in the data used (length of study periods), as well as to various climatological (differences between areas) or statistical reasons.

Above all, we noticed that Blanchet et al. (2009) actually focus on the new snow depth (snow depth accumulated in 1 day, referred to as "snowfall" in their article) while we focus on the snowfall (amount of solid precipitation in 1 day). Therefore, we believe that the elevation gradient of Blanchet et al. (2009) that we found in paragraph [31] of Gaume et al. (2013) are erroneous. Indeed, the gradient of new snow depth is reported in kg m$^{-2}$ / 100 m in Gaume et al. (2013), while it should have been reported in cm / 100 cm. Therefore, we will remove the reference to Blanchet et al. (2009).

Minor comments:

Title: I'd suggest to extend the title with «based on reanalysis» as the used snowfall data are not measured

We prefer not to change the title so that it remains a short and concise description of this study. We already indicate in the Abstract that our study is based on a reanalysis. Furthermore, the snowfall data are strongly based on measurements, since the SAFRAN reanalysis includes precipitation in-situ data in its analysis scheme (in contrast to other popular global reanalysis products such as ERA-Interim/ERA5 reanalyses). We fear that adding "based on reanalysis" could be misunderstood as "not representative of real-world trends".

Table 1: Why is Faranda et al (2020) not listed?

Faranda (2020) indeed focused on extreme snowfall (and we cite this work in the Discussion section). However, Table 1 focuses on the elevation-dependency of trends in extreme snowfall. To the best of our knowledge, Faranda (2020) rather focused on the spatial distribution of trends, but did not mention/illustrate the dependence with the elevation.

S nd : Why the mean of maximum? It's just the annual maximum snowfall in N consecutive days.

This notation was chosen based on the Table 2 of Frei et al. (2018) where S1d denotes the mean maximum of 1-day snowfall.

L61: Does 600 m elevation mean the 300 m elevation band between 450 and 750 m? Please clarify?

For the SAFRAN reanalysis "600 m elevation" means precisely that the corresponding data is taken at 600 m of elevation.

L242: I don't find the 1.5 kg m-2/100 m in Blanchet et al. (2009)?

The elevation gradient for the location and scale parameters of Blanchet et al. (2009) have been found in paragraph [31] of Gaume et al. (2013). However, Gaume et al. (2013) do not mention how they obtained these values. Gaume et al. (2013) report that Blanchet et al. (2009) focus on snowfall measured in kg m$^{-2}$ while they focus on "new snow depth" measured in cm (which is called "snowfall" in this article). Therefore, since Blanchet et al. (2009) did not study elevation gradient for snowfall, we will remove the reference to this article.

L243ff: This study is about annual maximum snowfall, which is only at low elevation a concern in regard to infrastructure. Avalanche defense structures or settlements in the mountains are endangered by annual maximum snow depth and not one snowfall event. Please rephrase!

We agree that settlements are rarely endangered by one snowfall event. Therefore, we will reformulate this section 5.3, and will add that "*extreme snow events are often triggered by one snowfall event but depend on other factors such as accumulated snow or wind.*" It is also true that individual snowfall events are particularly relevant, especially due to the fact that snow removal procedures are applied on top of buildings, so that for critical buildings there is generally no season long accumulation of snow on their roof.

L325: How is a "good" fit defined in numbers? What did you do for those cases, where no good fit could be obtained?

For a quantitative evaluation of the goodness of fit, we rely on the Anderson–Darling statistical test, which is the most powerful test for the Gumbel distribution (Abidin et al., 2012). This test assesses if the residuals follow a standard Gumbel distribution (see the Appendix A of our article for a definition of these residuals). For every selected model, the p-value of this test was computed for each elevation inside a range of elevation. On the Figure below, we observe that most tests are not rejected (with a 5% significance level) which quantitatively justify that the fits are generally "good".

[Figure]

Legend: Distribution of p-values for the Anderson-Darling test. For every selected model, a p-value was computed for each elevation inside a range of elevation.

In the case where no good fit could be obtained, we would have excluded the corresponding time series. However, in practice with the Q-Q plot we did not observe any fit to exclude. This adequate goodness of fit might be due to the fact that we consider 8 simple models (Tab. 2) that can cover a wide range of behavior.

L251-252: Is there a specific reason you distinguish here below 1000 and above 3000 m, in contrast to the separation below and above 2000 m in the abstract and the conclusion?

Our analysis shows that the contrast is mainly between below and above 2000 m for the past trends. In the sentence L251-252, we speculate about projected trends. Assuming that our past trends continue into the future, and following the literature (Tab. 1), it seems reasonable to state that it is likely that extreme snowfall will, in the near future, decrease below 1000 m, and increase above 3000 m. However, for elevations between 1000 m and 3000 m, results vary in the literature (Tab. 1).

L254-262: It seems strange that you write in these lines about increase in "extreme precipitation", "high latitude" and "moderate extreme snowfall", which all have so far not been analyzed in this study.

We agree that this part is unnecessarily speculative. In the revised manuscript, this section will be mostly deleted. Some of these elements have been added at the end of the perspective section.

 This might be true for moderate extreme snowfall but not for the annual maximum.

We did not understand this suggestion. Between L266-274, we indeed talk about extreme snowfall but not about annual maximum. In our manuscript, extreme snowfall stands for 100-year return level, while moderate extreme snowfall stands for the mean annual maxima.

We moved this sentence to paragraph 5.3 that starts with L243ff.

**References**

Abidin, N. Z., Adam, M. B., & Midi, H. (2012). The Goodness-of-fit Test for Gumbel Distribution: A Comparative Study. *Matematika*, *28*(1), 35–48. Retrieved from https://doi.org/10.11113/matematika.v28.n.313

Blanchet, J., Marty, C., & Lehning, M. (2009). Extreme value statistics of snowfall in the Swiss Alpine region. *Water Resources Research*, *45*(5). https://doi.org/10.1029/2009WR007916

Cabrera, A. T., Heras, M. De, Cabrera, C., & Heras, A. M. De. (2012). the Time Variable in the Calculation of Building Structures . How, 1–6. Retrieved from http://oa.upm.es/22914/1/INVE_MEM_2012_152534.pdf

Faranda, D. (2020). An attempt to explain recent trends in European snowfall extremes. *Weather Clim. Dynam.*, 445–458. Retrieved from https://doi.org/10.5194/wcd-1-445-2020

Frei, P., Kotlarski, S., Liniger, M. A., & Schar, C. (2018). Future snowfall in the Alps: projections based on the EURO-CORDEX regional climate models. *The Cryosphere*, *12*(1), 1–24. https://doi.org/10.3929/ETHZ-B-000233992

Gaume, J., Eckert, N., Chambon, G., Naaim, M., & Bel, L. (2013). Mapping extreme snowfalls in the French Alps using max-stable processes. *Water Resources Research*, *49*(2), 1079–1098. Retrieved from http://doi.wiley.com/10.1002/wrcr.20083

---

## Author Response (AR2)

**Point by point reply to the Interactive comments of Referee #3**

Dear Editors,

We thank the referees for their thorough reviews and for the numerous suggestions that greatly helped us to improve the manuscript.

Following the comments of Referee #3, we have:

- added to the discussion a sentence indicating that extreme snowfall has not yet been directly evaluated.

- corrected some notations in Table 1.

- added an Appendix section that illustrates the seasons when the annual maxima of daily snowfall occurred.

Following the recent publication of Blanchet al. 2021:

- We added at the line L296 a reference to this article: "We also observe for the period 1958-2017 that 20-year return level of winter precipitation have decreased in the north of the French Alps and have slightly increased or remained the same in the south (Fig. 8 of Blanchet et al. 2021)."

We hope that our revised manuscript will be found suitable for publication in "The Cryosphere"

Yours sincerely,

On behalf of the co-authors

Erwan Le Roux

We thank Reviewer #3 for his/her useful questions and comments on our manuscript. Please find below detailed feedback to individual comments and questions.

**Minor comments**

1. As requested some sentences regarding the evaluation of SAFRAN reanalysis have been added to chapter 2. However, it should be explicitly mentioned (maybe in the Discussion chapter) that extreme snowfall from SAFRAN reanalysis has so far not been evaluated.

We added the following sentence to the Discussion section "For future works, we note that a direct evaluation of extreme snowfall could help better pinpointing the locations where return levels might be biased".

2. Similarly, one or two sentences regarding the linear dependence of the location parameter (in contrast to Blanchet et al. 2009) should be added to the Discussion chapter. This maybe not only be "due to various differences in the data used" but also due to the fact that they analyzed the depth of snowfall (in cm) with corresponding settling and not water equivalent of snow fall (in mm).

The fact that they analyzed the depth of snowfall (in cm) while we focus on the snow water equivalent of daily snowfall (in mm) is included in the "various differences in the data used". We believe that we cannot conclude anything from the comparison with Blanchet et al. (2009) due to the numerous differences in data used. Therefore, we think that adding additional sentences regarding the linear dependence of the location parameter (in contrast to Blanchet et al. 2009) will only be misleading for the reader.

3. I would suggest to add the already produced plot and corresponding text about the months when the annual maxima of snowfall occurred (part of answer to reviewer #2) in the appendix, since the information is definitely interesting for some applications.

In the Discussion section, we replaced "We observe that at 2500 m (and at all elevations) that changes of 100-year return levels of winter precipitation, which may generate heavy snowfall show the same contrasted pattern than 100-year return levels of snowfall" with "We focus on winter precipitation because winter is the season where most annual maxima of daily snowfall occur below 3000 m (Appendix E). We observe that at 2500 m (and at all elevations) that changes of 100-year return levels of winter precipitation show the same contrasted pattern than 100-year return levels of snowfall".

In Appendix E, we illustrate the seasons when the annual maxima of daily snowfall occurred. We added the following text to this Appendix: "In Figure E1, we study the seasons when the annual maxima of daily snowfall occurred. For the elevation range 1 (below 1000 m) and for the elevation range 2 (between 1000 m and 2000 m), we observe that the annual maxima mainly occurred (>60%) between December and February, i.e. the coldest part of the snow season. For the elevation range 3 (between 2000 m and 3000 m) more than 40% of maxima occurred in winter, while slightly less than 30% occurred in autumn and in spring. For the elevation range 4 (above 3000 m) the seasons of occurrence are more spread, even if we observe that more than 40% of maxima occurred in autumn. In conclusion, we find that below 3000 m, most annual maxima of daily snowfall occur in winter, while above 3000 m they mostly occur in autumn. "

4. Table 1: "Mean of maximum" makes still no sense, because in contrast to the source of the term (Frei et al. (2018)) the authors analysis is not based on monthly values. Please just use "annual maximum snowfall in N consecutive days.

We replaced "mean of maximum" by "annual maximum", and removed the word "annual" in the column "Indicator" of Table 1.

**References**

Blanchet, J., Marty, C., & Lehning, M. (2009). Extreme value statistics of snowfall in the Swiss Alpine region. *Water Resources Research*, *45*(5). https://doi.org/10.1029/2009WR007916

Blanchet, J., Blanc, A., & Creutin, J.-D. (2021). Explaining recent trends in extreme precipitation in the Southwestern Alps by changes in atmospheric influences. *Weather and Climate Extremes*, 100356. https://doi.org/10.1016/j.wace.2021.100356

Frei, P., Kotlarski, S., Liniger, M. A., & Schar, C. (2018). Future snowfall in the Alps: projections based on the EURO-CORDEX regional climate models. *The Cryosphere*, *12*(1), 1–24. https://doi.org/10.3929/ETHZ-B-000233992